# Influence of Plasma Electrolytic Oxidation of Cast Al-Si Alloys on Their Phase Composition and Abrasive Wear Resistance

Mykhailo Student [1],*, Iryna Pohrelyuk [1], Juozas Padgurskas [2], Volodymyr Posuvailo [1],*, Volodymyr Hvozdets'kyi [1], Khrystyna Zadorozhna [1], Halyna Chumalo [1], Halyna Veselivska [1], Ihor Kovalchuk [1] and Andrii Kychma [3]

[1] Department of Material Science Bases of Surface Engineering, Karpenko Physico-Mechanical Institute of the National Academy of Sciences of Ukraine, 79060 Lviv, Ukraine; gvosdetcki@gmail.com (V.H.); 880988@ukr.net (K.Z.)

[2] Department of Mechanical, Energy and Biotechnology Engineering, Vytautas Magnus University, 44248 Kaunas, Lithuania

[3] Department of Technical Mechanics and Dynamics of Machines, Institute of Mechanical Engineering and Transport, Lviv Polytechnic National University, 79013 Lviv, Ukraine

* Correspondence: student-m-m@ipm.lviv.ua (M.S.); posuvaylo@ipm.lviv.ua (V.P.); +380-0969721972 (M.S.)

**Abstract:** The microhardness and abrasive wear resistance of cast Al-Si alloys after plasma-electrolytic oxidation (PEO) in a weakly alkaline basic electrolyte (3 g/L KOH + 2 g/L Na$_2$SiO$_3$), as well as with the addition of H$_2$O$_2$, were determined. X-ray analysis showed that the PEO layer comprises two oxide phases, namely $\alpha$-Al$_2$O$_3$ and $\gamma$-Al$_2$O$_3$, as well as sillimanite -Al$_2$O$_3 \cdot$ SiO$_2$ and a small percentage of mullite -3 Al$_2$O$_3 \cdot$ 2SiO$_2$. Silicon is present in the structure of the oxide layer, and its percentage is greater than that of the alloys in their initial state. It has been shown that the characteristics of PEO layers on AK9 and AK12 silumins synthesized in an electrolyte of basic composition increase (microhardness up to 900–1000 HV and abrasive wear resistance by 14–57 times). The formation of PEO layers in the base electrolyte with the addition of 3 g/L of hydrogen peroxide intensifies the synthesis process and promotes the formation of high-temperature oxide phases (in particular, corundum). The abrasive wear resistance of both silumins with PEO layers synthesized in such an electrolyte increases by 30–70%.

**Keywords:** plasma electrolytic oxidation; cast Al-Si alloys; PEO layers; abrasive wear resistance

## 1. Introduction

It is already difficult to find an industry where aluminum alloys are not used, from microelectronics to aviation to heavy metallurgy. This is due to its excellent physical and mechanical properties (low density and melting point, manufacturability, and ease of machining). Aluminum is one of the most abundant elements in the earth's crust, and its production cost is low. The physical and chemical properties of aluminum make it possible to consider it one of the most promising materials of the future. Aluminum-silicon alloys (Al-Si alloys), known as silumins, are the most common cast aluminum alloys used in mechanical engineering. Low melting points, excellent casting properties, and low cost of manufacturing parts characterized these alloys.

Siluminous materials are used to replace cast iron if the operating conditions exclude abrasive wear from the elements. However, in industrial equipment and mechanical engineering (in particular, on tractors, combines, and cars), large pulleys are used, which are often operated in the presence of abrasive particles in the air. Replacing cast iron pulleys with silumin ones significantly reduces their weight and axle loads, vibration levels, and energy consumption of machines. Aluminum casting is used to manufacture the internal combustion engine pistons, hydraulic cylinder rods, and brake discs. Parts made of cast aluminum alloys have a low cost. They can also operate under conditions of abrasive

wear. Despite the low cost of parts made of cast aluminum alloys, their use in mechanical engineering is limited by their low serviceability under abrasive wear conditions. Therefore, for such machine parts as pulleys, cylinders, pistons, guide planes, rails, rollers, elements of spinning and weaving equipment, impellers of compressors and pumps working with abrasive liquids, rolls, dies, and parts of extrusion equipment, it is necessary to ensure high abrasive wear resistance under working loads. To ensure the wear resistance and durability required for such parts, methods are being actively developed to improve the wear resistance of light alloys [1–4].

In order to protect aluminum alloys, chromium plating and hard anodizing technologies are most often used [5–9]. These technologies have an advantage over others due to their lower energy consumption during the formation of protective coatings. The thickness of the protective layers obtained using these methods is 100–200 μm and their microhardness is 800–1000 HV. However, these methods have a significant disadvantage since they use environmentally hazardous electrolytes (especially in the chromium plating process). In addition, coatings obtained by the hard anodizing method on pure aluminum have significantly better properties than those on cast alloys. The ban on chromium plating prompts researchers to actively search for and develop new methods of forming wear-resistant layers on aluminum alloys and silumins. These include laser surface treatment, surface remelting, and the introduction of graphite and sillumin nanocomposites [10–13]. The coatings obtained by these methods have a small thickness of 50–100 μm and a microhardness of 120–300 HV. The formation of coatings by electron doping provides them with high microhardness (up to 2.34 GPa) and wear resistance, which is 18–20 times higher than that of the matrix [14].

Recently, the plasma electrolytic oxidation (PEO) technology of aluminum alloys has been rapidly developing. This technology is one of the most promising for applying protective coatings to products made of light structural alloys. This was confirmed in a number of recent publications [15–20] devoted to the study of the mechanisms and kinetics of the growth of PEO coatings on the surfaces of light structural alloys formed under various technological process conditions. In the same work, the properties and features of the structures of the obtained coatings were also analyzed. Particular attention is also paid by researchers to combining the possibilities of various methods of surface hardening [21].

The advantages of PEO technology are as follows: (1) the small production areas for the implementation of the process and its low duration (surfaces of elements do not need careful preparation before the process); (2) the ability to obtain anti-corrosion coatings with high mechanical characteristics (hardness, wear resistance, adhesion to a substrate, low coefficient of friction); and (3) sufficiently high environmental friendliness.

PEO layers synthesized on deformed aluminum alloys have better physical, mechanical, and tribological characteristics compared to layers formed on cast aluminum alloys (silumins). Significant disadvantages of the synthesis of PEO layers on silumins are the low rate of their synthesis (0.5–1 μm per minute), small thickness (up to 80–100 μm), microhardness (700–900 HV), and high energy consumption for their formation (a current density of 30–40 A/dm$^2$). However, the mechanisms of synthesis of PEO layers on cast aluminum alloys are not fully understood. Therefore, the synthesis of PEO layers is an important area of research.

The aim of the work is to synthesize PEO layers on cast aluminum alloys AK9 and AK12 in electrolytes of different compositions and study their microstructure to determine the microhardness and abrasive wear resistance (with a fixed abrasive).

## 2. Materials and Methods

Oxide ceramic layers were synthesized on two cast Al-Si alloys of the following composition, wt%: AK9 (88.8 Al; 9.9 Si; 0.5 Cu; 0.5 Ti; 0.2 Mg; 0.1 Mn) and AK12 (85.2 Al; 13.5 Si; 0.7 Fe; 0.6 Cu). Their analogs (in accordance with the ANSI USA nomenclature) are the A03600 and A04130 ANSI USA alloys, respectively.

The dimensions of the samples were 50 Ч 30 Ч 5 mm³. Before synthesis, all samples were polished and washed in distilled water and ethyl alcohol. The cathode-anode mode of formation of PEO layers with a change in the polarity of the applied voltage and a frequency of 50 Hz was used. The duration of the cathode and anode pulses was 6 ms. The PEO layers were formed within 2 h using an IMPELOM setup. Cathodic and anodic current densities were $j_c/j_a = 15/15$ A/dm². An aqueous solution of KOH (3 g/L) and $Na_2SiO_3$ (2 g/L) was used as the electrolyte in the basic composition. In addition, different amounts of $H_2O_2$ (3, 5, and 7 g/L) were also added to the basic electrolyte.

X-ray phase diffraction (XRD) analysis of PEO layers was performed using a DRON 3.0 M diffractometer ($CuK_6$-radiation) in a step mode (step 0.05°, exposure at a point of 10–15 s, 2 И = 20°–90°, cathode voltage 30 kV, current-10 mA). The crystal structure was refined by the Rietveld technique using the FullPfof software for X-ray data [22–25]. The analysis of cross sections of the PEO layers was carried out by a ZEISS EVO-40XVP scanning electron microscopy (SEM) equipped with an energy-dispersive X-ray (EDX) microanalysis and backscattered detector (BSD) system, the INCA Energy 350. The microhardness was measured using a PMT-3 microhardness tester under a load of 50 g (referring to ISO 6507–1:2005 for Vicker's hardness tests). The thicknesses of the PEO layers were determined using a CHY TG-05 thickness gauge with a measurement range of 0–300 μm. They were measured at different points (at least 10 times) on samples with an area of 30 × 50 mm², and then their average values were determined. This was sufficient to estimate the thickness of the PEO layers with an accuracy of ±10%.

An abrasive wear resistance test with the fixed abrasive was carried out using an abrasive disk (Figure 1) with a diameter of 150 mm and a width of 8 mm made of electro-corundum with a grain size of 250–315 μm. The disk rotation frequency was 160 rpm, and the load in the area of linear contact was 15 N. The abrasive wear resistance of samples was evaluated by weight loss using an electronic scale (KERN ABJ 220 4M) with an accuracy of $2 \times 10^{-4}$ g.

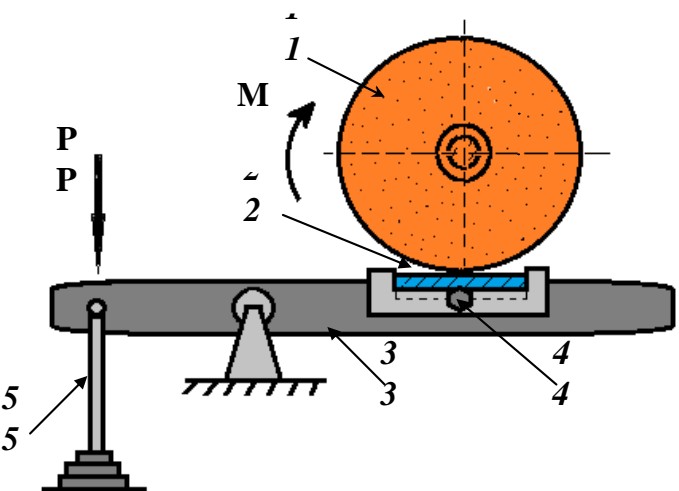

**Figure 1.** Schematic diagram of the device for abrasive wear testing of specimens with PEO layers synthesized on their surface: 1—abrasive wheel; 2—specimen; 3—lever transmitting force; 4—specimen fixation elements; 5—weights; P—load.

## 3. Results and Discussion

### 3.1. Morphology of the Surface of PEO Layers Synthesized on Cast Alloys AK9 and AK12 in Different Electrolytes

An analysis of the surface morphology of PEO layers obtained in an electrolyte of their basic composition and with the addition of hydrogen peroxide showed similar features for both alloys. In particular, the synthesis of PEO layers in the basic electrolyte contributed to the cracking of their surface layers (Figure 2a,b). As a result, cracks appeared along

the boundaries of the craters at the exit of the plasma discharge channels to the surface of the PEO layers. They formed a fairly developed network of damage on the surface of the PEO layers of both alloys. As regards the shape and dimensions of the plasma discharge channels in both alloys, they did not differ significantly in general. However, cracks on the surfaces of both alloys could increase the wear of PEO layers with such morphology.

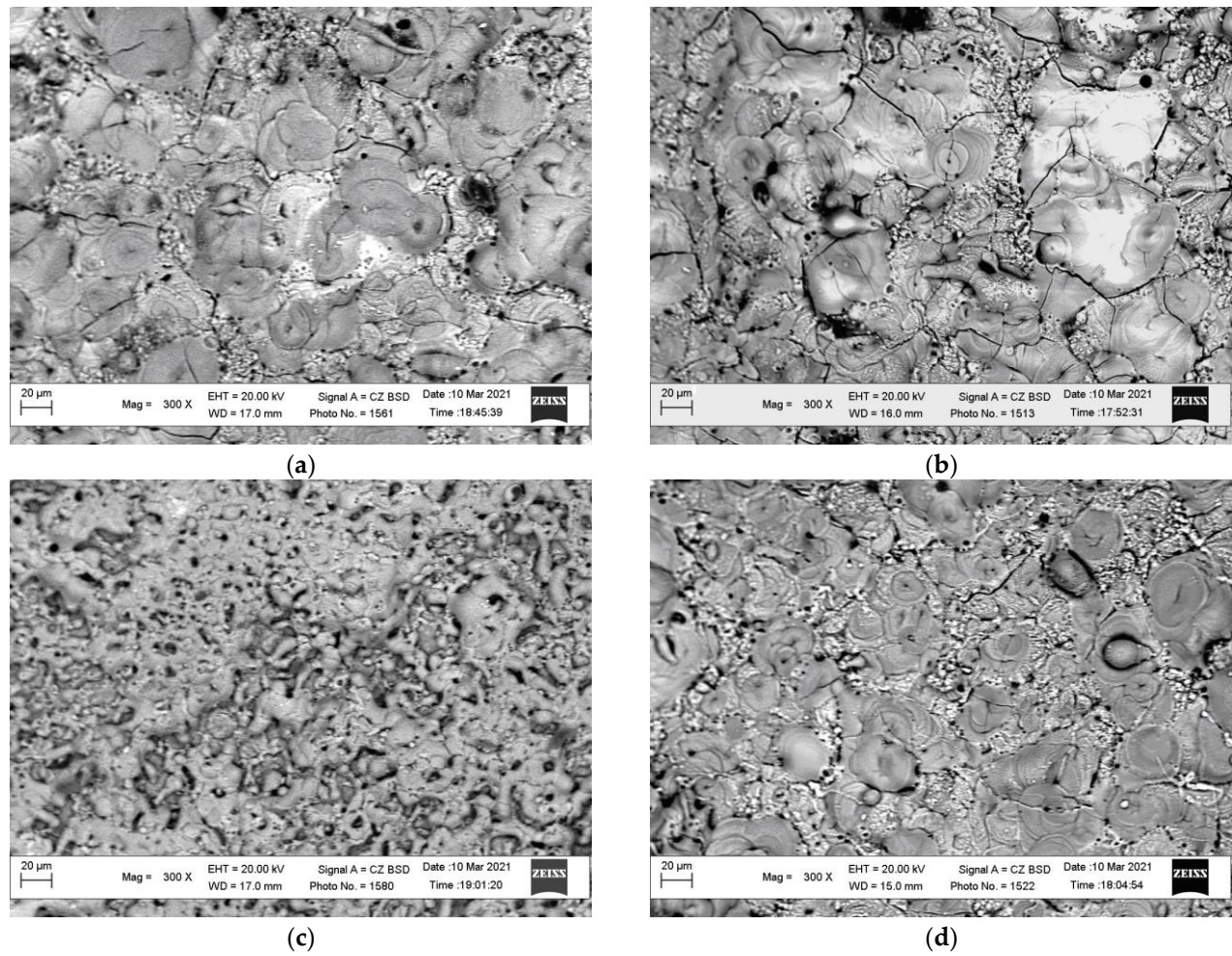

**Figure 2.** Morphology of the surface of PEO layers on cast alloys AK9 (**a**,**c**) and AK12 (**b**,**d**) synthesized in the base electrolyte (aqueous solution of 3 g/L KOH + 2.1 g/L Na$_2$SiO$_3$) (**a**,**b**) and with the addition to the base composition of hydrogen peroxide in the amount of 3 g/L (**c**,**d**).

Signs of cracking on the surface of the PEO layers of both alloys during their synthesis in an electrolyte with the addition of hydrogen peroxide to the base composition were practically not observed (Figure 2c,d). In this case, pores of different sizes became dominant defects. This was especially pronounced on the surface of the PEO layer in the AK9 alloy, on which the pore size reached 5 μm (Figure 2c). At the same time, smaller pores (up to 1 μm) were observed on the surface of the PEO layer on the AK12 alloy. The craters at the exit of the plasma discharge channels to the surface of the PEO layer were also much smaller in this case. This provided a fairly uniform surface morphology of the PEO layer on the AK12 alloy and its more uniform structure (Figure 2d). Although microcracks were practically not observed on the surface of the PEO layer on the AK12 alloy (in contrast to the case of synthesis in the electrolyte of the basic composition), a network of tracks from very small pores was observed around the conglomerates from craters. They were considered a factor in the heterogeneity of the surface morphology, which can facilitate wear.

### 3.2. Comparison of AK9 and AK12 Alloys and PEO Layers Synthesized on Their Surfaces Using SEM-EDX Images

In order to compare the composition of aluminum alloys (AK9 and AK12) and the PEO layers on their surfaces (synthesized in the electrolyte of the basic composition), an X-ray microanalysis was performed over the area (Figure 3). According to data obtained by recalculating the content of elements in the PEO layers (after excluding oxygen from consideration), the silicon to aluminum content ratios exceeded the values characteristic of the alloys themselves. Namely, these ratios are 0.16 and 0.18 for the AK9 alloy and the PEO layer on its surface, respectively. Whereas for the AK12 alloy and the PEO layer on its surface, these ratios are 0.18 and 0.23, respectively. An increase in the relative content of silicon in the PEO layers of both alloys (compared to the values inherent in the alloys themselves) is possible only in the case of its additional absorption during synthesis from an electrolyte containing $Na_2SiO_3$. This contributed to an increase in the amount of $Al_2O_3 \cdot nSiO_2$ compounds in the composition of the PEO layer.

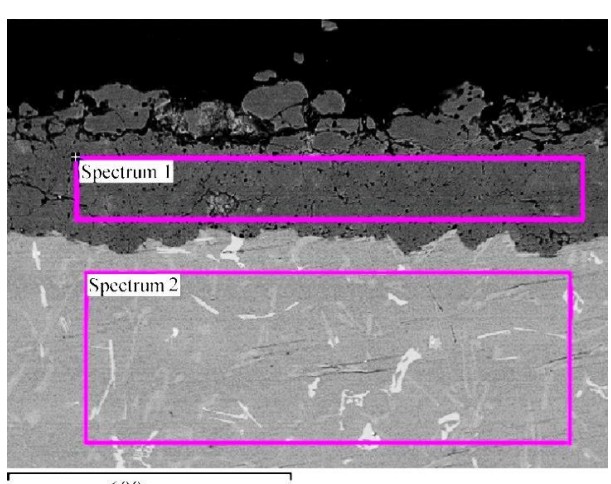

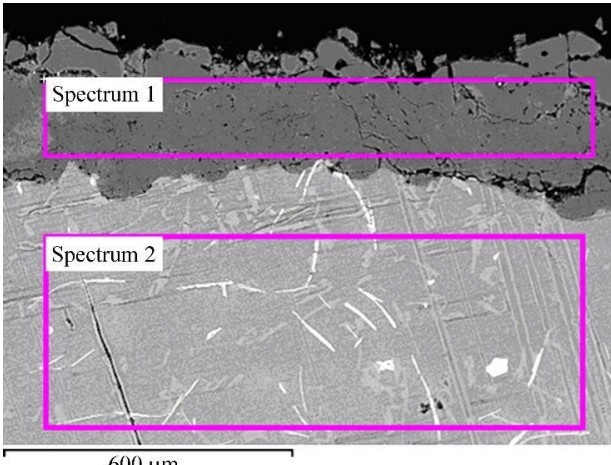

| Spectrum 1 (PEO leyer) | | |
| --- | --- | --- |
| **Element** | **Wt%** | **At. %** |
| O K | 43.25 | 56.45 |
| Al K | 49.63 | 38.41 |
| Si K | 6.36 | 4.73 |
| K K | 0.76 | 0.41 |
| Total | 100.00 | |

| Recalculation of the content of elements in the PEO layer, excluding oxygen | | |
| --- | --- | --- |
| **Element** | **Wt%** | **At. %** |
| Al K | 83.52 | 84.44 |
| Si K | 14.88 | 14.45 |
| K K | 1.60 | 1.12 |
| Total | 100.00 | |

| Spectrum 2 (substrate) | | |
| --- | --- | --- |
| **Element** | **Wt%** | **At. %** |
| Al K | 85.44 | 86.27 |
| Si K | 13.75 | 13.33 |
| Fe K | 0.81 | 0.40 |
| Total | 100.00 | |

**(a)**

| Spectrum 1 (PEO leyer) | | |
| --- | --- | --- |
| **Element** | **Wt%** | **At. %** |
| O K | 43.56 | 56.85 |
| Al K | 47.37 | 36.66 |
| Si K | 7.85 | 5.84 |
| K K | 1.22 | 0.65 |
| Total | 100.00 | |

| Recalculation of the content of elements in the PEO layer, excluding oxygen | | |
| --- | --- | --- |
| **Element** | **Wt%** | **At. %** |
| Al K | 79.39 | 80.60 |
| Si K | 18.06 | 17.61 |
| K K | 2.55 | 1.79 |
| Total | 100.00 | |

| Spectrum 2 (substrate) | | |
| --- | --- | --- |
| **Element** | **Wt%** | **At. %** |
| Al K | 84.41 | 85.15 |
| Si K | 15.06 | 14.59 |
| Fe K | 0.53 | 0.26 |
| Total | 100.00 | |

**(b)**

**Figure 3.** Results of X-ray microspectral analysis of the area of PEO layers synthesized in the base electrolyte on the surface of two cast aluminum alloys AK9 (**a**) and AK12 (**b**).

### 3.3. Structure and Phase Composition of the PEO Layer Synthesized in Different Electrolytes on AK9 Alloy

Figure 4 shows the structure of the oxide-ceramic layer formed on the surface of the AK9 aluminum alloy. First, the uneven growth of the front of the PEO layer into the depth of the substrate is obvious. This causes a significant change in the thickness of the PEO layer at the front of its synthesis. It is clear that this feature is more noticeable at high resolution. Secondly, a rather high homogeneity of the inner part of the PEO layer (near the substrate) and an increased density of pores and cracks in its outer layer were noted. Thirdly, a significant amount of silicon crystals found in the substrate structure practically did not show up in the structure of the PEO layer. This indicates that, during the synthesis of the PEO layer, the initial silicon crystals (observed in the structure of the AK9 alloy) were mainly transformed into oxide phases.

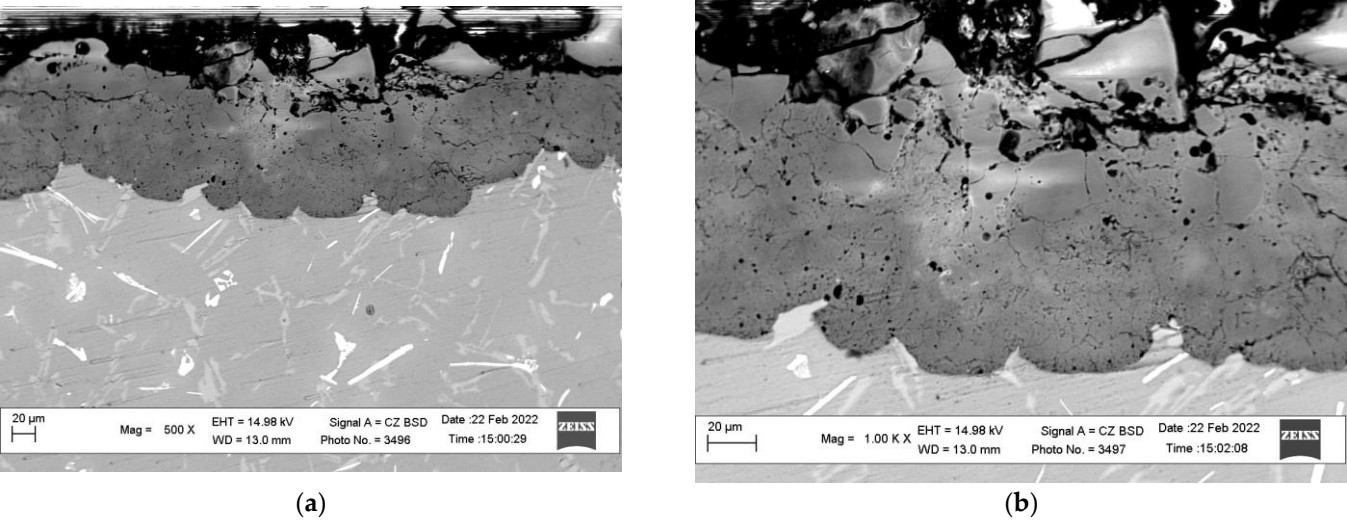

(**a**)          (**b**)

**Figure 4.** Structure of the PEO layer synthesized on the AK9 alloy in a weakly alkaline basic electrolyte (3 g/L KOH + 2 g/L Na$_2$SiO$_3$), shown at low (**a**) and higher (**b**) (instead different) resolutions.

Unlike Al–Cu–Mg alloys, the PEO layer on the AK9 alloy grows unevenly (nonuniformly) into the matrix [24]. In contrast to Al–Cu–Mg alloys, in which the PEO layer propagates into the substrate with a more or less even front [26], on the AK9 alloy, the PEO layer is synthesized nonuniformly along its front. It was assumed that the growth of the PEO layer slowed down when silicon crystals were encountered on the way of its propagation deep into the substrate. This is confirmed by the image of the synthesis front of the PEO layer deep into the substrate, which is stopped by a successfully oriented chain of silicon crystals encountered in its path (Figure 5). Silicon has a high melting point (T-1683 K) and boiling point (T-2628 K) [27]. It is a semiconductor whose electrical conductivity depends on impurities and temperature. During the synthesis of PEO layers, high-temperature plasma discharges can increase the substrate temperature up to 300 °C. This contributes to an increase in the electrical conductivity of silicon crystals and ensures the passage of electric current through them.

Obviously, during the synthesis of the PEO layer on the AK9 alloy, a high temperature is reached in the discharge channels and around them, which leads to the melting of silicon crystals. In addition, volatile compounds, such as SiO and SiH, can form during the interaction of silicon with electrolyte components in the discharge channels. At high temperatures (more than 2100 °C), these compounds can evaporate and pass both into the electrolyte and into the oxide ceramic layer. Such processes lead to an increase in the number of pores and, consequently, a decrease in the microhardness of the PEO layer.

The XRD patterns of the PEO layer synthesized on the AK9 alloy in 3 g/L KOH + 2 g/L Na$_2$SiO$_3$ electrolytes are shown in Figure 6.

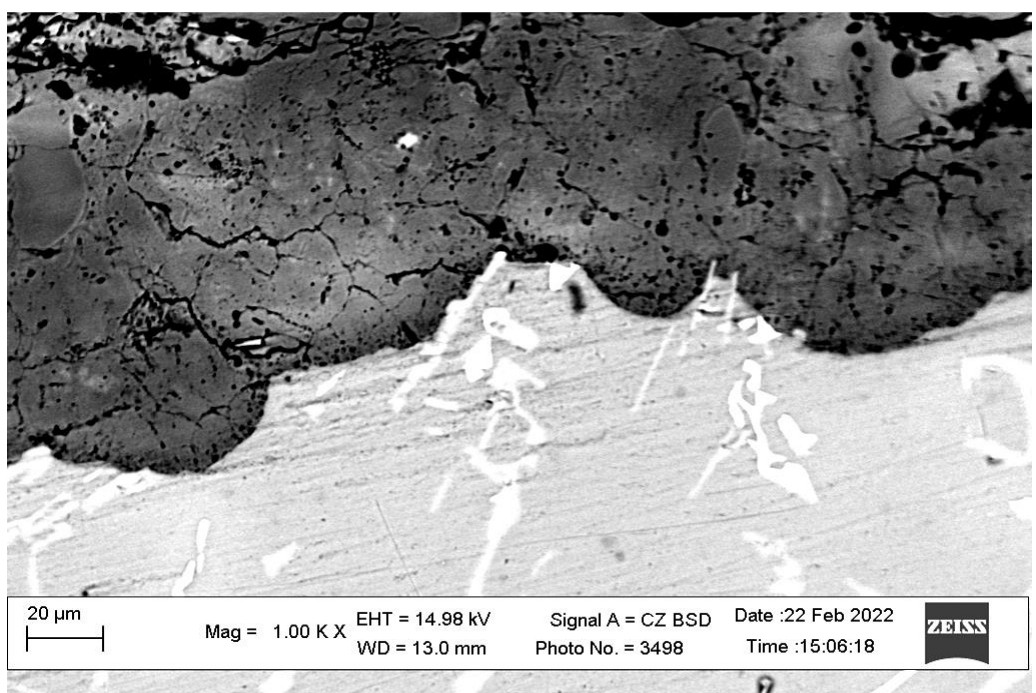

**Figure 5.** Visualization of the deceleration of the synthesis front of the PEO layer on the AK9 aluminum alloy in the electrolyte of the base composition, which occurred on the silicon crystals encountered on its way.

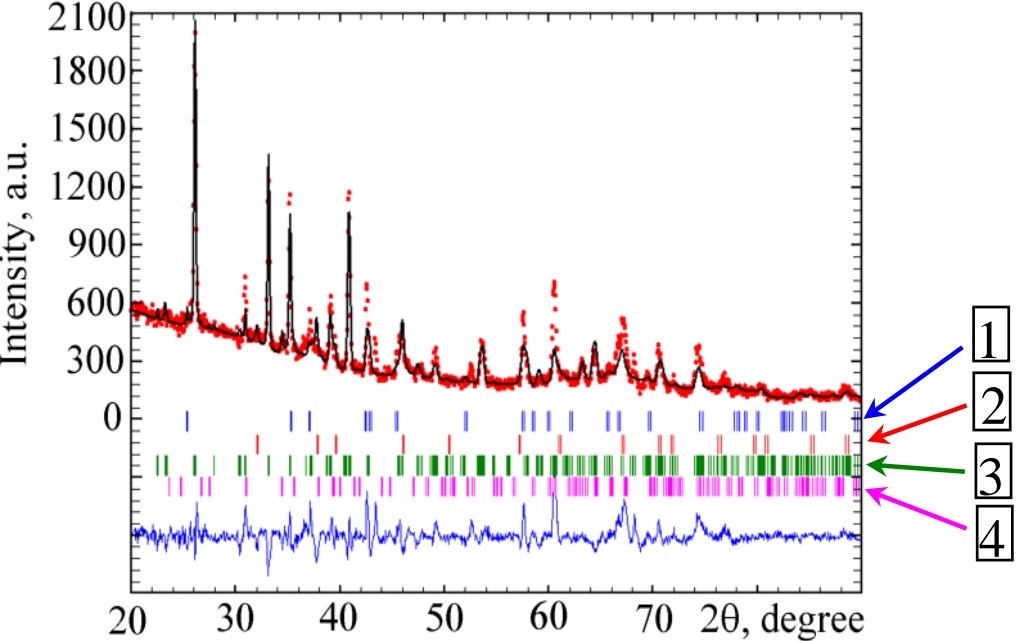

**Figure 6.** The observed (red dots at top), calculated (lines in the middle) and differential (blue line at the bottom) XRD patterns (in CuKα-radiation) of the PEO layer synthesized on the AK9 alloy in a weakly alkaline basic electrolyte of 3 g/L KOH + 2 g/L $Na_2SiO_3$ composition. The numbers indicate the reflections corresponding to (1) α-$Al_2O_3$, (2) γ-$Al_2O_3$, (3) $SiO_2$ and (4) $Al_2O_3 \cdot SiO_2$ phases.

All compounds in the composition of the PEO layer were detected by XRD method (DRON-3.0, in Cu-K$_\alpha$ radiation). The obtained XRD data were refined with the use of the FullProf program [25]. The refined crystallographic parameters of all phases and their contents are presented in Table 1.

**Table 1.** Crystallographic parameters and quantitative content of phases in the PEO layer synthesized on the AK9 alloy in a slightly alkaline electrolyte of 3 g/L KOH + 2 g/L Na$_2$SiO$_3$ composition.

| Phase | $a$, Å | $b$, Å | $c$, Å | PG | wt% |
|---|---|---|---|---|---|
| $\alpha$-Al$_2$O$_3$ | 4.889 (8) | 4.889 (8) | 12.71 (2) | $R$-$3C$ | 10 |
| $\gamma$-Al$_2$O$_3$ | 7.893 (2) | 7.893 (2) | 7.893 (2) | $Fd$-$3m$ | 25 |
| SiO$_2$ | 5.006 (6) | 5.006 (6) | 5.505 (2) | $P3_12_1$ | 5 |
| Al$_2$O$_3$·SiO$_2$ | 7.610 (9) | 7.665 (4) | 5.781 (2) | $Pbnm$ | 60 |

The oxide ceramic layer mainly consists of ɼ-Al$_2$O$_3$, Al$_2$O$_3$·SiO$_2$, ƃ-Al$_2$O$_3$ and a small amount of SiO$_2$.

Some approaches were developed to increase the growth rate of the layer [28]. The addition of hydrogen peroxide to a weakly alkaline electrolyte of 3 g/L KOH + 2 g/L Na$_2$SiO$_3$ causes an increase in the temperature in the plasma discharge channels and an increase in the amount of oxygen and hydrogen in the discharge channels and their surroundings. This leads to both an increase in the rate of aluminum oxide formation and an increase in the amount of volatile SiO and SiH compounds [29]. This effect is manifested in the formation of pores and microcracks in the vicinity of silicon inclusions at the border between the PEO layer and the matrix. An additional factor leading to the formation of a network of microcracks is the rapid cooling of the newly formed aluminum oxide by the electrolyte (Figure 7).

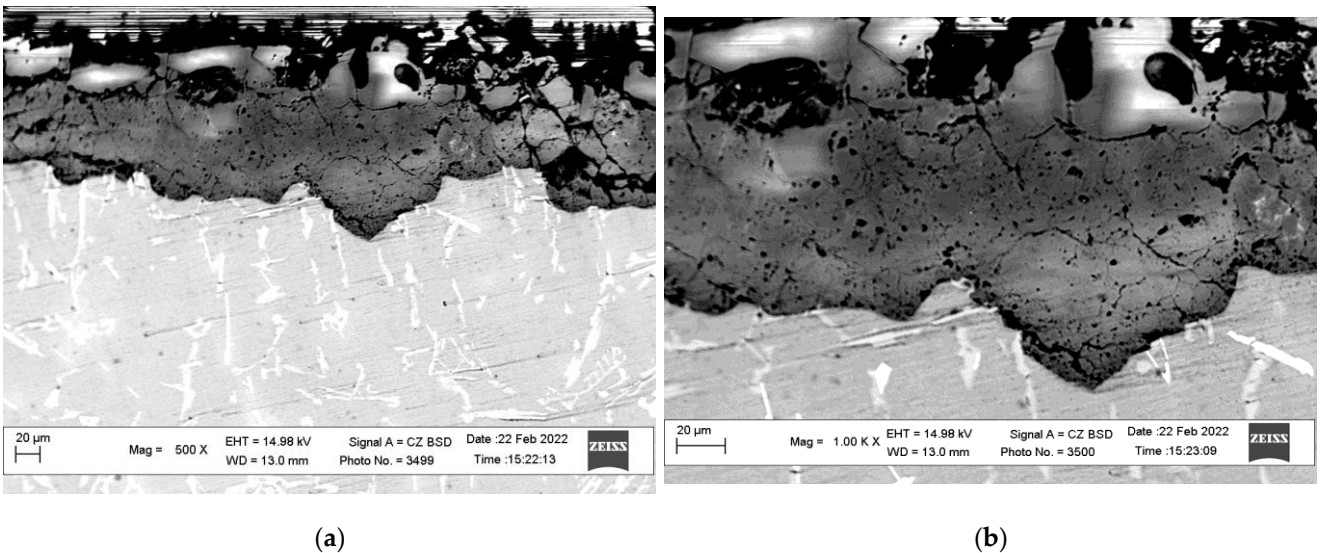

(**a**)                                                          (**b**)

**Figure 7.** Structure of a PEO layer at at low (**a**) and higher (**b**) (instead different) resolutions synthesized in an electrolyte of 3 g/L KOH + 2 g/L Na2SiO3 + 3 g/L H$_2$O$_2$ on the surface of an AK9 aluminum alloy.

The phase compositions of the PEO layer synthesized in 3 g/L KOH + 2 g/L Na$_2$SiO$_3$ electrolyte and in the same electrolyte with the addition of 3 g/L H$_2$O$_2$ are given in Tables 1 and 2 respectively. Hydrogen peroxide contributes to the synthesis of a larger amount of the high-temperature ƃ-Al$_2$O$_3$ and $\gamma$- Al$_2$O$_3$ phases in the PEO layer (Figure 8, Table 2). However, the main phases of the PEO layer synthesized on silumin AK9 are Al$_2$O$_3$·SiO$_2$ and $\gamma$-Al$_2$O$_3$. The X-ray pattern of the PEO layer synthesized on the AK9 alloy in a 3 g/L KOH + 2 g/L Na$_2$SiO$_3$ + 3 g/L H$_2$O$_2$ electrolyte is shown in Figure 8. The refined crystallographic parameters of all phases and their contents are presented in Table 2.

**Table 2.** Crystallographic parameters and quantitative content of phases in the PEO layer synthesized on the AK9 alloy in a weakly alkaline electrolyte of 3 g/L KOH + 2 g/L Na$_2$SiO$_3$ + 3 g/L H$_2$O$_2$ composition.

| Phase | *a*, Å | *b*, Å | *c*, Å | S.G. | wt% |
|---|---|---|---|---|---|
| α-Al$_2$O$_3$ | 4.749 (9) | 4.749 (9) | 12.969 (3) | *R-3C* | 10 |
| γ-Al$_2$O$_3$ | 7.896 (3) | 7.896 (3) | 7.896 (3) | *Fd-3m* | 40 |
| SiO$_2$ | 5.006 (6) | 5.006 (6) | 5.505 (2) | *P3$_1$2$_1$* | 10 |
| Al$_2$O$_3$·SiO$_2$ | 7.610 (9) | 7.665 (4) | 5.781 (2) | *Pbnm* | 40 |

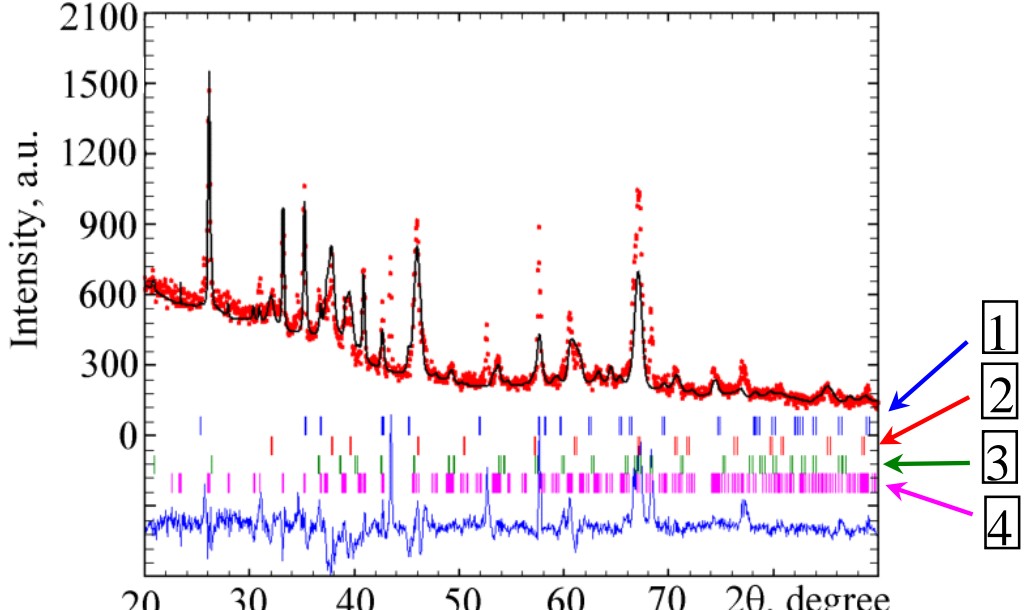

**Figure 8.** The observed (red dots at top), calculated (lines in the middle) and differential (blue line at the bottom) XRD patterns (in CuKα-radiation) of the PEO layer synthesized on the AK9 alloy in a weakly alkaline basic electrolyte of 3 g/L KOH + 2 g/L Na2SiO3 composition. The numbers indicate the reflections corresponding to (1) α-Al$_2$O$_3$, (2) γ-Al$_2$O$_3$, (3) SiO2 and (4) Al$_2$O$_3$·SiO2 phases.

A spectral analysis over the area with the distribution of elements in the substrate and PEO layer synthesized on the AK9 alloy in an electrolyte of 3 g/L KOH + 2 g/L Na$_2$SiO$_3$ is shown (Figure 9). It was found that the distribution of oxygen in the layers is uniform. The distributions of aluminum and silicon are non-uniform. An increased content of silicon was observed in areas of the PEO layer near silicon inclusions as well as in areas where silicon inclusions were encountered along the PEO layer synthesis front. Silicon inclusions at the boundary of the initial alloy and oxide ceramic slow down the growth of the PEO layer.

The distribution of elements in the cross-sectional area of the PEO layers obtained with different electrolytes is almost the same. The large cross-sectional area of the samples was analyzed. However, the content of silicon can differ significantly in small areas (Figure 9). At the same time, the phase composition of PEO layers synthesized in the electrolyte with the addition of H$_2$O$_2$ changed significantly. Synthesis in such an electrolyte changes the content of such phases in the PEO layer as γ-Al$_2$O$_3$, Al$_2$O$_3$·SiO$_2$, and SiO$_2$ (compare the data in Tables 1 and 2), which can affect its wear resistance. It depends on the distribution of silicon in the initial state of the alloy. The content of corundum in PEO layers synthesized in an electrolyte with the addition of H$_2$O$_2$ remained almost unchanged.

### 3.4. Structure and Phase Composition of PEO Layer Synthesized in Different Electrolytes on AK12 Alloy

The results of metallographic studies confirmed the negative effect of silicon crystals in the substrate structure (cast alloy AK12) on the advancement of the PEO synthesis front. During the synthesis of PEO layers on the AK12 alloy (as in the synthesis of the AK9 alloy),

silicon crystals in the substrate slow down the synthesis process at the interfaces between the silicon crystals and the matrix, and the propagation of their fronts into the depth of the substrate also slows down (Figure 10). The inhibitory role of crystals is clearly manifested when the crystals are favorably oriented with respect to the synthesis front of the PEO layer. This is the reason for the significant curvilinearity of the front of the synthesis process.

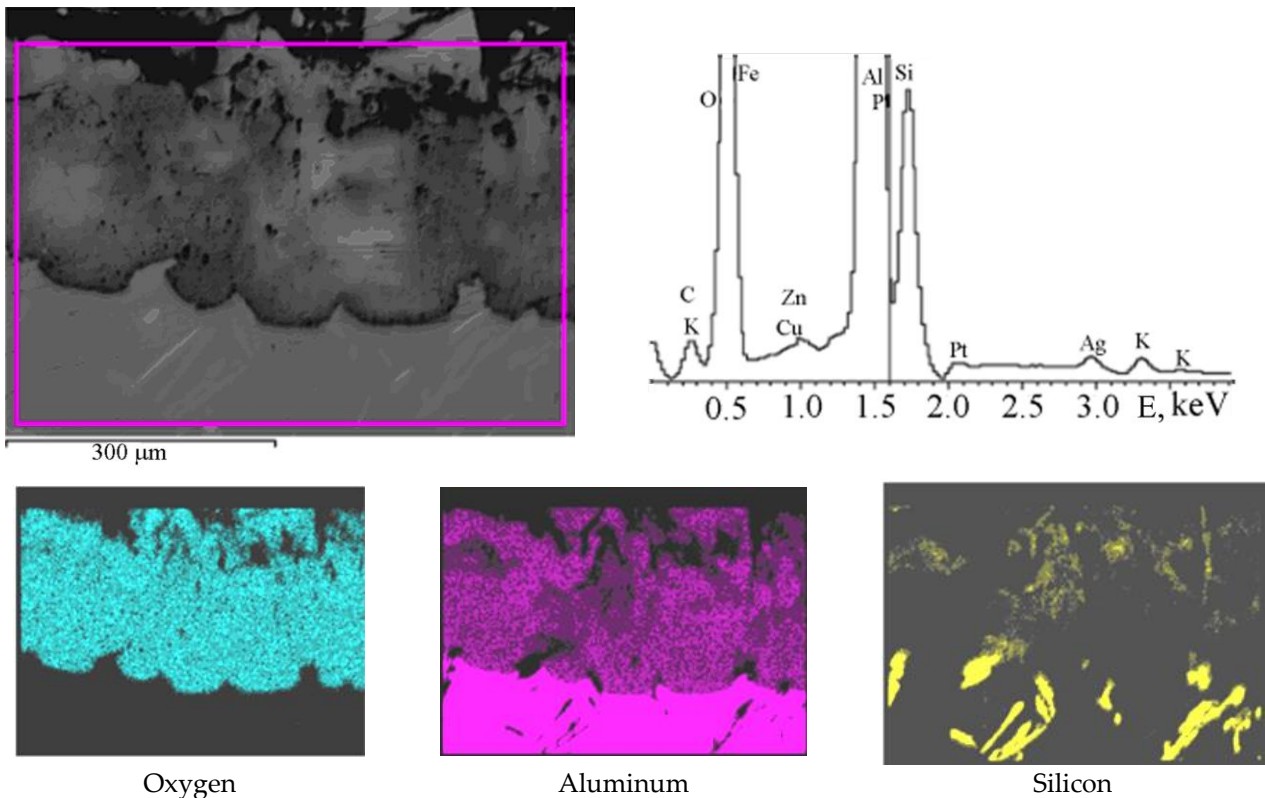

Oxygen                      Aluminum                      Silicon

**Figure 9.** Distribution of oxygen, aluminum and silicon over the cross-sectional area of the PEO layer on the AK9 alloy synthesized in 3 g/L KOH + 2 g/L $Na_2SiO_3$ electrolyte.

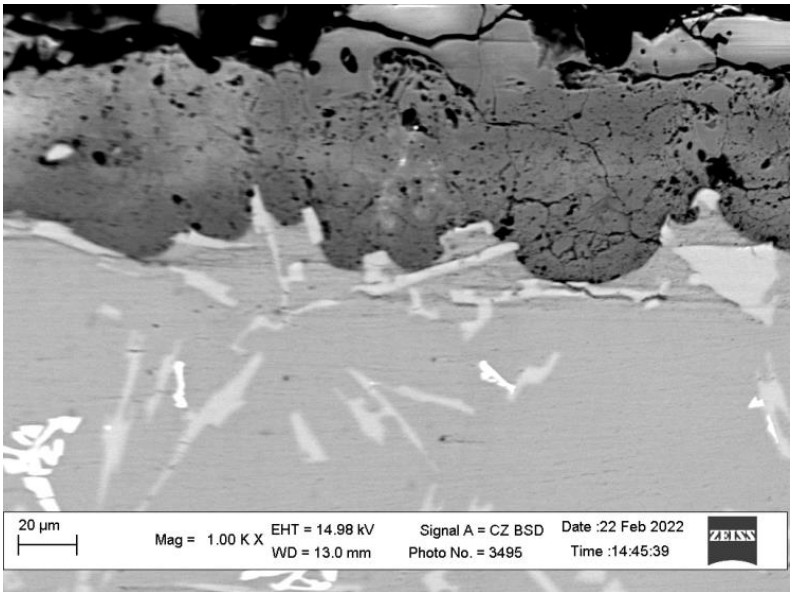

**Figure 10.** Structure of the PEO layer synthesized on AK12 alloy in 3 g/L KOH + 2 g/L $Na_2SiO_3$ electrolyte.

The XRD patterns of the PEO layer synthesized on the AK12 alloy in 3 g/L KOH + 2 g/L Na$_2$SiO$_3$ electrolytes are shown in Figure 11. The refined crystallographic parameters of all phases and their contents are presented in Table 3. The previous subsection also describes the results of the XRD phase analysis of the PEO layer synthesized on the AK9 alloy in 3 g/L KOH + 2 g/L Na$_2$SiO$_3$ electrolyte (Table 1).

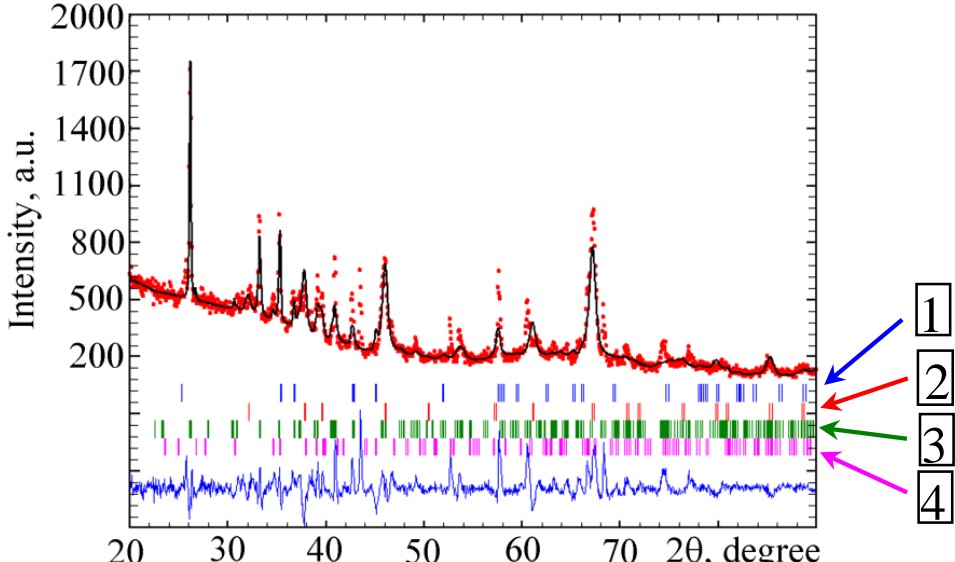

**Figure 11.** The observed (red dots at top), calculated (lines in the middle) and differential (blue line at the bottom) XRD patterns (in CuK$\alpha$-radiation) of the PEO layer synthesized on the AK12 alloy in a slightly alkaline electrolyte of 3 g/L KOH + 2 g/L Na2SiO3 composition. The numbers indicate the reflections corresponding to (1) $\alpha$-Al$_2$O$_3$, (2) $\gamma$-Al$_2$O$_3$, (3) SiO2 and (4) Al$_2$O$_3$·SiO2 phases.

**Table 3.** Crystallographic parameters and quantitative content of phases in the PEO layer synthesized on AK12 alloy in 3 g/L KOH + 2 g/L Na$_2$SiO$_3$ electrolyte.

| Phase | $a$, Å | $b$, Å | $c$, Å | S.G. | wt% |
|---|---|---|---|---|---|
| $\alpha$-Al$_2$O$_3$ | 4.893 (5) | 4.893 (5) | 12.653 (1) | *R-3C* | 6 |
| $\gamma$-Al$_2$O$_3$ | 7.877 (4) | 7.877 (4) | 7.877 (4) | *Fd-3m* | 25 |
| SiO$_2$ | 4.881 (2) | 4.881 (2) | 5.700 (1) | *P3$_1$2$_1$* | 4 |
| Al$_2$O$_3$·SiO$_2$ | 7.610 (9) | 7.665 (4) | 5.781 (2) | *Pbnm* | 65 |

The refined crystallographic parameters and the content of the phases determined by the X-ray diffraction method in the composition of the PEO layer synthesized on the AK12 alloy in the base electrolyte (3 g/L KOH + 2 g/L Na$_2$SiO$_3$) are presented in Table 3. Evidently, the content of the Al$_2$O$_3$·SiO$_2$ phase in the PEO layer on the AK12 alloy is higher than on the AK9 alloy (compare the data presented in Tables 1 and 3). This is a quite predictable result, since initially the silicon content in the composition of the AK12 alloy was higher than that of the AK9 alloy. Further, taking into account the retarding effect of silicon crystals on the advancement of the synthesis front deep into the substrate, one could also expect a thinner PEO layer on the AK12 alloy. Moreover, the smaller amount of ɑ-Al$_2$O$_3$ and r-Al$_2$O$_3$ in the PEO layer of the AK-12 alloy also agrees with the results for silicon-containing phases. In the PEO layer synthesized on the AK12 alloy in 3 g/L KOH + 2 g/L Na$_2$SiO$_3$ electrolyte, the amount of the Al$_2$O$_3$·SiO$_2$ phase is increased (Figure 11; Table 3). The refined crystallographic parameters of all phases and their contents are presented in Table 3.

The addition of hydrogen peroxide to 3 g/L KOH + 2 g/L Na$_2$SiO$_3$ electrolytes leads to an increase in the number of microcracks and a decrease in the size of pores in the PEO layer on the AK12 alloy (Figure 1d). The distribution of elements over the area of

the PEO layer on the AK12 alloy, synthesized in the electrolyte, records the accumulation of silicon crystals, which prevent the growth of the PEO layer deep into the aluminum matrix (Figure 9). Similar to the AK9 alloy, the accumulation of silicon crystals near the synthesis front of the PEO layer in the AK12 alloy also prevents its penetration deep into the aluminum matrix (Figure 10). And since the silicon content in the AK12 alloy is higher than that in AK9, the effect of silicon crystals on the synthesis process retardation will be stronger, and, consequently, the thickness of the PEO layer in AK12 will be less.

The XRD patterns of the PEO layer synthesized on the AK12 alloy in a (3 g/L KOH + 2 g/L $Na_2SiO_3$ + 3g/L $H_2O_2$) electrolyte are shown in Figure 12. The refined crystallographic parameters of all phases and their contents are shown in Table 4.

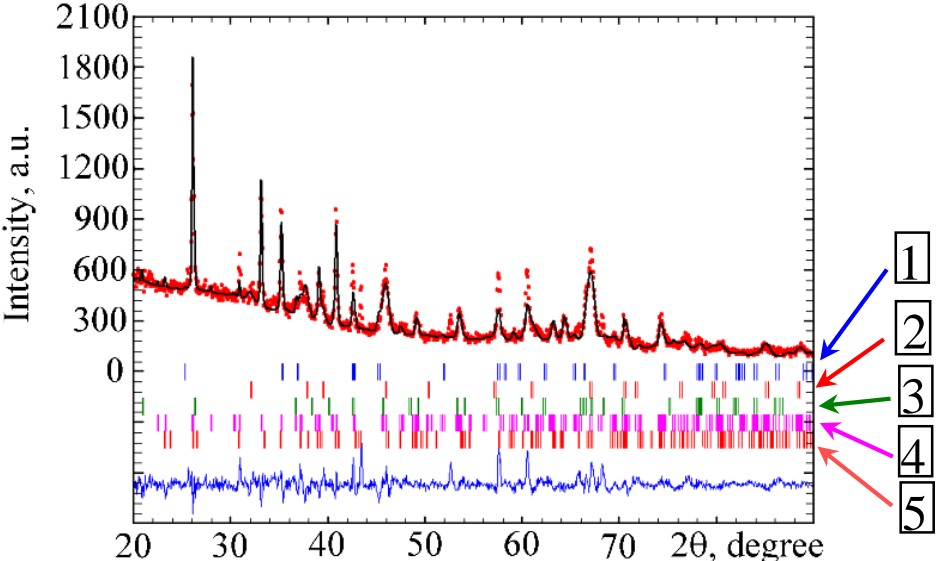

**Figure 12.** The observed (red dots at top), calculated (lines in the middle) and differential (blue line at the bottom) XRD patterns (in CuKα-radiation) of the PEO layer synthesized on the AK12 alloy in a weakly alkaline electrolyte of 3 g/L KOH + 2 g/L Na2SiO3 + 3g/L $H_2O_2$ composition. The numbers indicate the reflections corresponding to (1) $\alpha$-$Al_2O_3$, (2) $\gamma$-$Al_2O_3$, (3) $SiO_2$, (4) $Al_2O_3 \cdot SiO_2$ and (5) $3Al_2O_3 \cdot 2SiO_2$ phases.

**Table 4.** Crystallographic parameters and quantitative content of phases in the PEO layer synthesized on AK12 alloy in a weakly alkaline electrolyte of 3g/L KOH + 2 g/L Na2SiO3 + 3g/L $H_2O_2$ composition.

| Phase | $a$, Å | $b$, Å | $c$, Å | S.G. | wt% |
|---|---|---|---|---|---|
| $\alpha$-$Al_2O_3$ | 4.884 (5) | 4.884 (5) | 12.72 (1) | *R-3C* | 10 |
| $\gamma$-$Al_2O_3$ | 7.894 (9) | 7.894 (9) | 7.894 (9) | *Fd-3m* | 35 |
| $SiO_2$ | 4.885 (2) | 4.885 (2) | 5.700 (7) | $P3_12_1$ | 20 |
| $Al_2O_3 \cdot SiO_2$ | 7.610 (9) | 7.665 (4) | 5.781 (2) | *Pbnm* | 32 |
| $3Al_2O_3 \cdot 2SiO_2$ | 7.531 (1) | 7.208 (2) | 2.888 (9) | *Pbam* | 3 |

The addition of 3g/L $H_2O_2$ to the electrolyte significantly reduces the quantitative content of the $Al_2O_3 \cdot SiO_2$ phase (sillimanite), and a small percentage of $3Al_2O_3 \cdot 2SiO_2$ phase (3.2 wt%) is revealed in the structure of the PEO layer on the AK12 alloy (Table 4). At the same time, the amount of r-$Al_2O_3$ and silicon oxide $SiO_2$ increases.

The number of pores in the structure of the synthesized PEO layer on the AK12 alloy increased with the addition of hydrogen peroxide to the base electrolyte (3 g/L KOH + 2 g/L $Na_2SiO_3$). This was visible on the surface of the PEO layer (Figure 1d) and confirmed in its cross section (Figure 13). However, in both cases (both on the surface and in the cross section), the pores in the structure were smaller than in the case of synthesis in the basic

electrolyte. The PEO layer in the AK12 alloy (in the addition of $H_2O_2$ to the electrolyte composition) propagates into the depth of the substrate nonuniformly (the same as in the case of the synthesis of the PEO layer in the base electrolyte), since silicon crystals slow down its propagation (Figure 13b). Consequently, silicon crystals in cast aluminum alloys limit the synthesis of PEO layers, and the addition of $H_2O_2$ to the base electrolyte has little effect on eliminating this disadvantage.

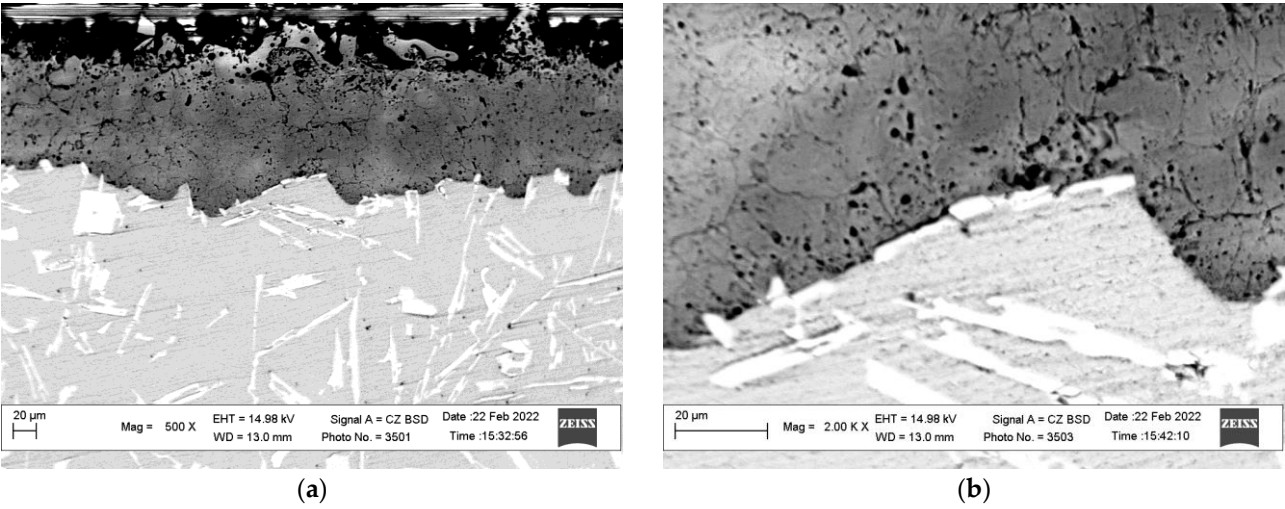

(**a**)  (**b**)

**Figure 13.** Structure of a PEO layer at low (**a**) and higher (**b**) (instead different) resolutions synthesized on the AK12 alloy in a slightly alkaline electrolyte of 3 g/L KOH + 2 g/L $Na_2SiO_3$ + 3 g/L $H_2O_2$ composition.

The high content of silicon in silumins contributes to the formation of high-temperature phases in PEO coatings, in particular α-$Al_2O_3$, mullite $3Al_2O_3·2SiO_2$ and especially sillimanite-$Al_2O_3·SiO_2$. The amount of sillimanite increases due to the inclusion of the volatile compounds SiO and SiH in the oxide ceramic coating.

The dense arrangement of silicon crystals near the oxidation front exhausts the adjacent matrix for the content of aluminum in it, and the synthesis of aluminum-silicon-containing oxides becomes more complicated. Therefore, the content of pure silicon oxide in the PEO layer on the AK12 alloy increases.

A spectral analysis of the distribution of elements in the PEO layers on the AK12 alloy is shown in Figure 14. It has been established that the distribution of oxygen and aluminum in the layers is more or less uniform. The distribution of silicon is non-uniform. The silicon content increased in the areas of the oxide ceramic PEO layer near the silicon inclusions as well as in the areas where there were silicon inclusions during the growth of the oxide ceramic layer along the PEO layer synthesis front (Figure 14). Silicon inclusions at the interface of the original alloy substrate and oxide ceramics slow down the growth of the oxide ceramic PEO layer. The amount of silicon around the silicon inclusions is greater than in the PEO layers on the AK9 alloy.

The addition of hydrogen peroxide to the electrolyte does not lead to significant differences in the distribution of elements in the oxide ceramic layer on the AK12 alloy.

Despite a rather noticeable change in the phase composition of the PEO layer synthesized in electrolyte with the addition of hydrogen peroxide, the distribution of elements over the area of the PEO layer on the AK12 alloy did not undergo significant changes.

### 3.5. Microhardness of PEO Layers, Synthesized on Silumins

The microhardness of PEO layers significantly depends on the alloying of aluminum alloys. PEO layers synthesized on silumins have significantly lower microhardness than PEO layers synthesized on Al–Cu and Al–Cu–Mg alloys. Thus, the PEO layer on the D16 alloy (analog of AA2024 ANSI USA), synthesized in the basic alkaline electrolyte-(3 g/L

KOH + 2 g/L Na$_2$SiO$_3$), has a microhardness of 1600–1900 HV [28]. The PEO layer on the AMg6 alloy (analog of AA 5056 ANSI USA) synthesized in the basic alkaline electrolyte (3 g/L KOH + 2 g/L Na$_2$SiO$_3$) has the microhardness of 1200–1600 HV [30]. Whereas the PEO layers on AK9 and AK12 silumins, synthesized in the same electrolyte, have significantly lower microhardness. Their values do not exceed 850–960 HV (Figure 15).

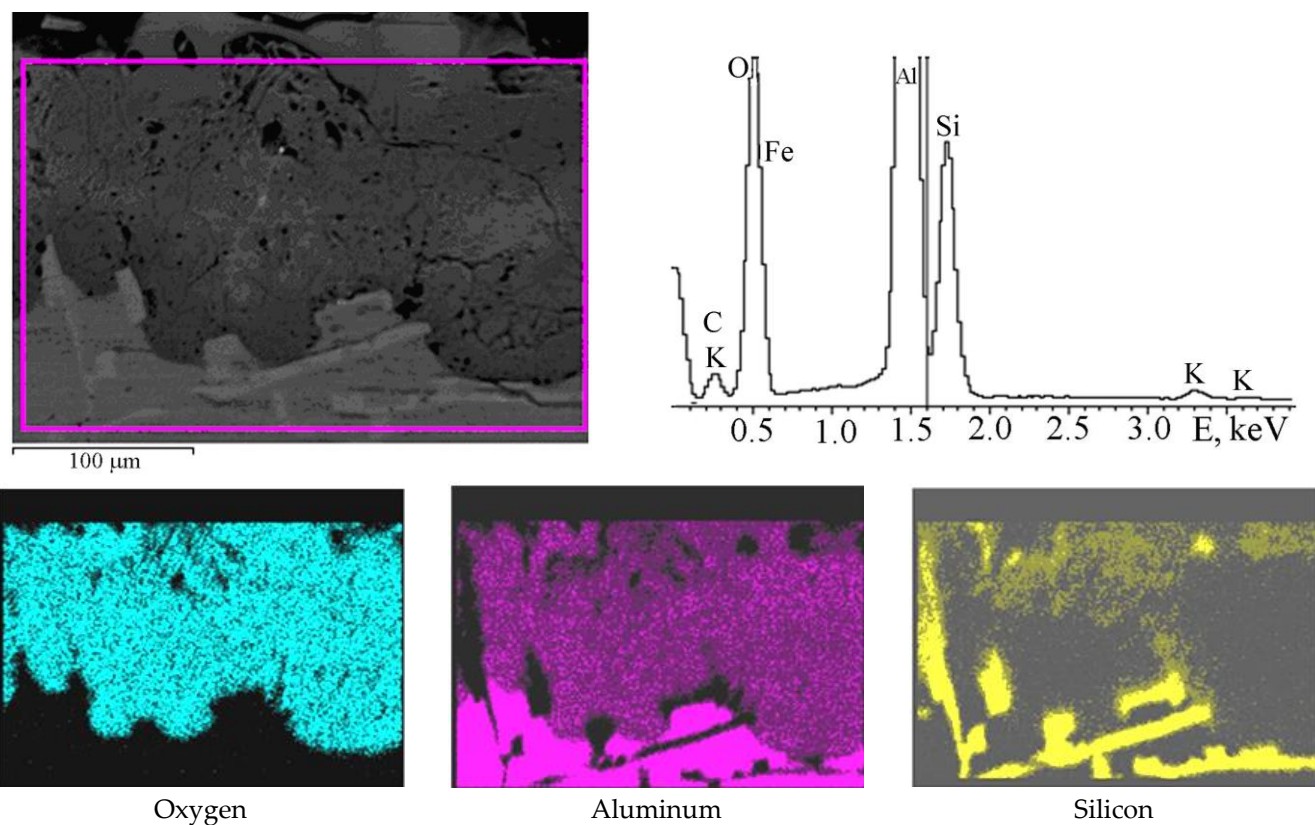

**Figure 14.** Distribution of oxygen, aluminum and silicon over the cross-sectional area of the PEO layer on the AK12 alloy synthesized in 3 g/L KOH + 2 g/L Na$_2$SiO$_3$ electrolyte.

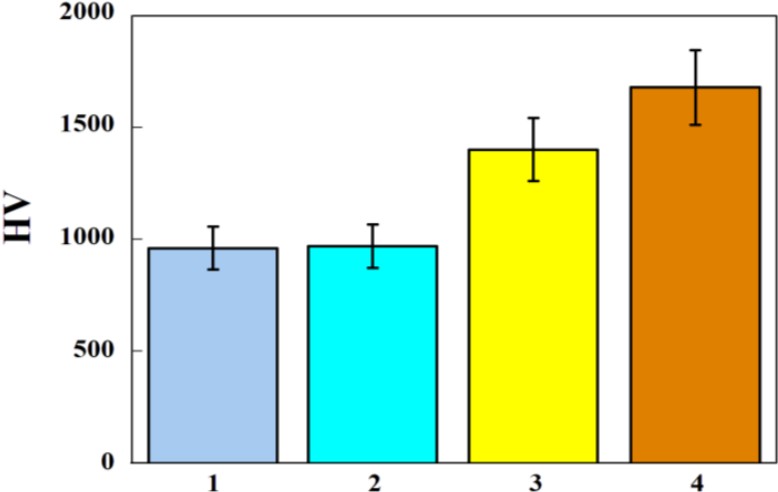

**Figure 15.** Influence of the alloy composition of the aluminum alloys (1—AK12; 2—AK9; 3—AMg6; 4—D16) on the microhardness of PEO layers synthesized in 3 g/L KOH + 2 g/L Na$_2$SiO$_3$ electrolyte.: 1—AK12; 2—AK9; 3—AMg6; 4—D16.

In order to increase the microhardness of PEO layers on silumins, a strong oxidizing agent (hydrogen peroxide) was added to the electrolyte for the synthesis of PEO layers [28]. The formation of aluminum oxide in plasma discharges is described by the following formulas:

$$2Al + 3O = Al_2O_3 \tag{1}$$

$$2Al + 3OH = Al_2O_3 + 3H \tag{2}$$

$$4Al + 3O_2 = 2Al_2O_3 \tag{3}$$

$$2Al + 3OH^- = Al_2O_3 + 3H + 3\ e^- \tag{4}$$

Based on the law of the active masses, it is possible to increase the yield of aluminum oxide according to the reactions (1)–(4) by increasing concentrations of reagents, in particular O, $O_2$, OH, and $OH^-$ [31].

Active electrolysis of water and hydrogen peroxide occurs in the plasma-discharge channels, and, as a result, during PEO, they easily decompound with the release of oxygen.

As a result of the thermal decomposition of hydrogen peroxide in the plasma discharge channels, a larger amount of oxygen is formed. This effect leads to an increase in the coating thickness. The thickness of PEO layers on AK9 and AK12 alloys in electrolytes with different hydrogen peroxide contents is shown in Figure 16. The addition of hydrogen peroxide to the electrolyte leads to an increase in the thickness of the PEO layers, both on the AK9 alloy and on the AK12 alloy. The maximum thickness of 240–300 μm is observed on the PEO layers formed on the AK9 alloy at concentrations of 3 and 5 g/L $H_2O_2$ (Figure 16, white bars). A further increase in the concentration of peroxide leads to a decrease in the thickness of the coating. The thickness of the PEO layers formed on the AK12 alloy is somewhat smaller. This alloy has a larger amount of silicon, which, due to its properties, slows down the growth of the coating (Figure 16, dark bars). An increase in the concentration of hydrogen peroxide in the electrolyte to 7 g/L leads to some reduction in the coating thickness. This is explained by a significant increase in the pH of the electrolyte and the predominance of aluminum oxide dissolution processes over its synthesis processes.

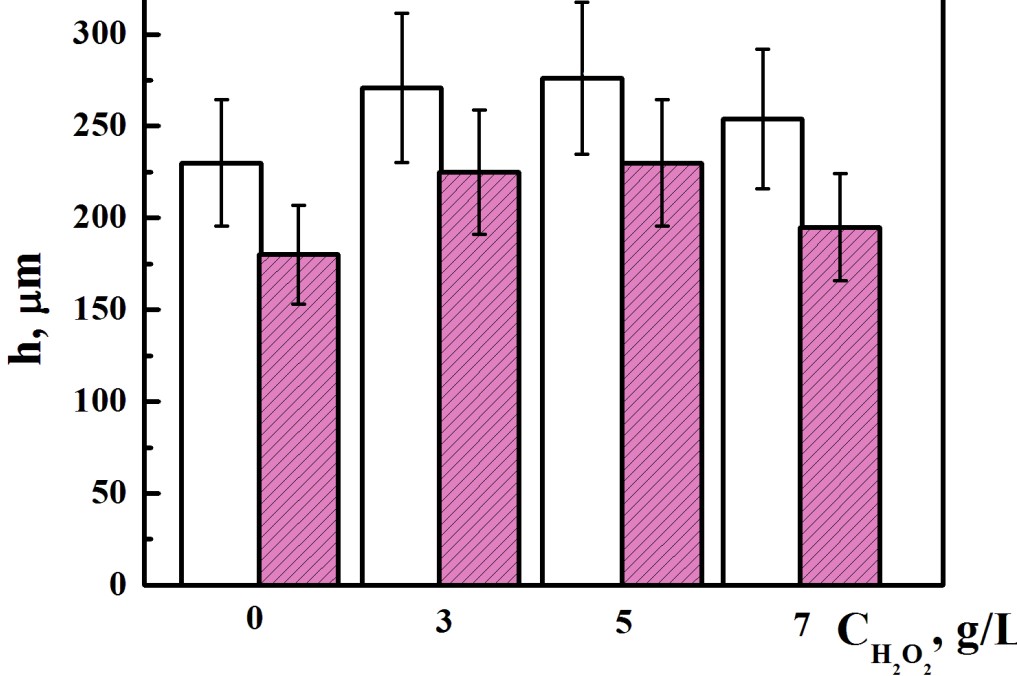

**Figure 16.** Effect of the amount of hydrogen peroxide $C_{H_2O_2}$ in the main electrolyte on the thickness of PEO layers h on alloys AK-9 (white bars) and AK12 (dark bars).

The increase in thickness leads to the synthesis of more corundum in the PEO layer. The addition of 3 g/L $H_2O_2$ to the base electrolyte (3 g/L KOH + 2 g/L $Na_2SiO_3$) significantly increases the microhardness of the PEO layers on silumins. Thus, the microhardness of the PEO layer on the AK9 alloy increased from 960 to 1050 HV, and the microhardness of the PEO layer on the AK12 alloy increased from 850 to 1000 HV. It should be noted that separate measurements of the microhardness of PEO layers on the AK9 alloy synthesized in electrolyte with the addition of 3 g/L $H_2O_2$ have a value of 2000 HV. This corresponds to the microhardness of corundum. PEO layers on the AK12 alloy do not have such microhardness. Although the phase analysis of the PEO layers on the AK12 alloy indicates the presence of corundum in the layer. This may be due to the influence of SiO and SiH on the formation of a large number of small micropores in the PEO layers, which was clearly manifested on the surface of the PEO layer on the AK12 alloy (Figure 2d). The rather high microhardness of the synthesized PEO layers on both aluminum alloys makes it possible to predict their rather high abrasive wear resistance.

### 3.6. Abrasive Wear Resistance of PEO Layers, Synthesized on Silumins

The abrasive wear resistance in tests with a fixed abrasive is one of the most common test methods used to study the properties of materials. The abrasive wear resistance of PEO layers synthesized on AK9 and AK12 alloys was studied at their contact with a SM-2 electrocorundum disk on a 7K15 ceramic bond with a hardness of 1800–1900 HV.

The obtained results indicate that the oxide ceramic layers synthesized on AK9 alloy in the basic electrolyte with the addition of 3 g/L of hydrogen peroxide have the highest wear resistance (Figure 17). The oxide ceramic layers synthesized on AK12 alloy in electrolytes with different $H_2O_2$ contents have lower abrasive wear resistance. The lowest abrasive wear resistance for PEO layers synthesized on AK12 alloy was observed in 3 g/L KOH + 2 g/L $Na_2SiO_3$ electrolytes. An increase in the amount of hydrogen peroxide in the electrolyte leads to an increase in the wear resistance the oxide ceramic layers. In contrast to the AK9 alloy, the layers synthesized in a 3 g/L KOH + 2 g/L $Na_2SiO_3$ + 5 g/L $H_2O_2$ electrolyte have the highest wear resistance. A further increase in hydrogen peroxide concentration in the electrolyte leads to a decrease in wear resistance. PEO layers synthesized in an electrolyte with 7 g/L $H_2O_2$ have a lower growth rate and lower corundum content [31]. This happens due to the increase in pH of the electrolyte and the dissolution of the oxide layer.

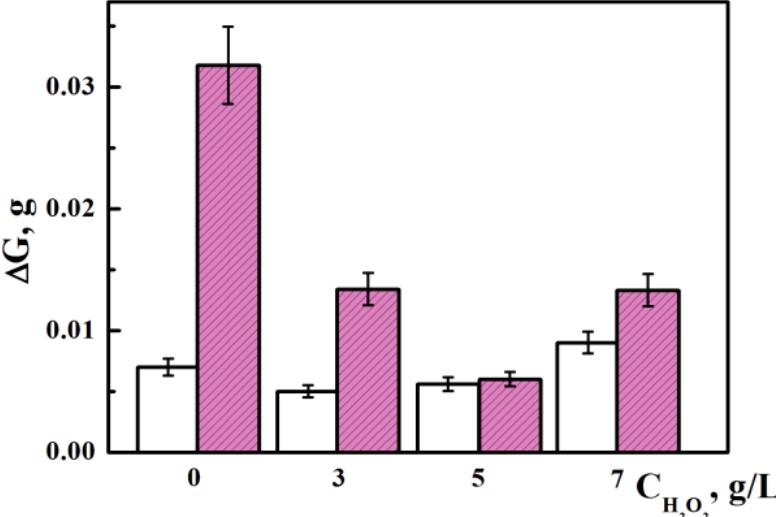

**Figure 17.** Influence of the content of hydrogen peroxide $C_{H_2O_2}$ in the basic electrolyte on the weight loss $\Delta G$ of PEO layers on the surfaces of the AK-9 (white bars) and AK12 (dark bars) alloys during wear resistance tests with a fixed abrasive.

The weight loss of samples from AK9 and AK12 alloys in the initial state and with PEO layers synthesized in electrolytes 3 g/L KOH + 2 g/L $Na_2SiO_3$ and 3 g/L KOH + 2 g/L $Na_2SiO_3$ + 3 g/L $H_2O_2$ are compared after abrasive wear resistance tests (Figure 18). The wear resistance of AK9 alloy in its initial state is higher than that of AK12 alloy. This is explained by the higher content and larger sizes of solid silicon crystals in the AK12 alloy. During the tests, they break away from the alloy and act as an abrasive. This leads to increased weight loss of the sample with the PEO layer during the test. The oxide ceramic layers synthesized on both alloys in a 3 g/L KOH + 2 g/L $Na_2SiO_3$ electrolyte significantly increase their wear resistance. The abrasive wear resistance of the PEO layer synthesized on the AK9 alloy is almost 80 times higher than that of the alloy in the initial state. The abrasive wear resistance of the PEO layer on the AK12 alloy is about 20 times higher than that of the alloy in its initial state. Consequently, the PEO layers on the AK9 alloy have an order of magnitude higher wear resistance than the AK12 alloy. This effect can be explained by the smaller number of silicon inclusions and their smaller sizes, and the more uniform structure of the synthesized oxide ceramic layers. The network of microcracks observed on the surface of the PEO layers on both alloys did not negatively affect their abrasive wear resistance. Perhaps this is due to the relaxation of tensile stresses in PEO layers due to the formation of microcracks in them. The addition of hydrogen peroxide to the electrolyte leads to an increase in the corundum content of the PEO layers [31] and an increase in their wear resistance. The abrasive wear resistance of the PEO layer on the AK9 alloy increased by 30% and on the AK12 alloy by 70% after adding hydrogen peroxide to the base electrolyte. It is possible that this effect is due to smaller pores on the surface of the PEO layer on the AK12 alloy synthesized in an electrolyte with the addition of $H_2O_2$. In addition, an increase in the abrasive wear resistance can occur as a result of the grinding of the wheel oxide phases $\gamma$-$Al_2O_3$ and $SiO_2$ with low microhardness (as a consequence of contact with corundum abrasive) and the formation of a finely dispersed powder that acts as a solid lubricant.

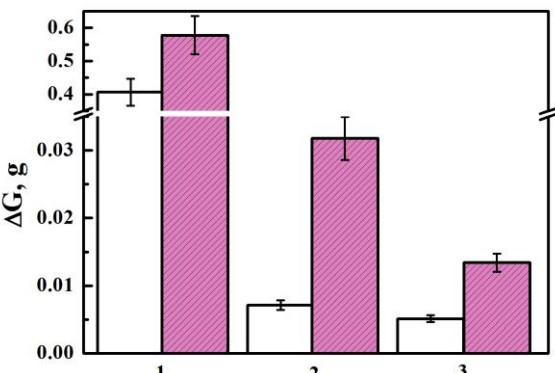

**Figure 18.** Weight loss $\Delta G$ of samples with PEO layers on alloys AK9 (white bars) and AK12 (dark bars) after testing for abrasive wear resistance with a fixed abrasive: 1—alloys in the initial state; 2—alloys with PEO layers synthesized in the base electrolyte (3 g/L KOH + 2 g/L $Na_2SiO_3$); 3—alloys with PEO layers synthesized in the base electrolyte with the addition of 3 g/L $H_2O_2$.

During the synthesis of coatings, the optimal current densities and the time of synthesis of PEO layers were determined. All coatings were synthesized at a current density of 15 A/dm$^2$ for 2 h. This made it possible to obtain a larger amount of corundum [32]. An increase in current density and time leads to a significant increase in the power of plasma channels. The SiO and SiH compounds are formed in the plasma discharge channels and exit the coating. This leads to the formation of layers with a large number of pores and cracks. The structure of such coatings is often given in the literature [33–37]. On the basis of the research carried out, simple electrolytes have been developed that can significantly increase the microhardness and wear resistance of AK9 and AK12 alloys. The results presented in the article are compared with those known in the literature. A particularly

interesting method is the synthesis of PEO layers with TiC nanoparticles inside them [36]. The successful synthesis of PEO layers with nanosized TiC inclusions was reported in a number of works [36–38]. Although the microhardness of such coatings is higher, namely 1300–1600 HV, their wear resistance is lower than that obtained in this study. This can be explained by the effect of solid TiC nanoparticles embedded in the oxide matrix. These small particles act as an abrasive and further reduce the wear resistance of the synthesized layers. At the same time, in our case, the hardest phase in the synthesized PEO layers was $\alpha$-$Al_2O_3$ corundum. The rest phases contained in the PEO layers can act as a solid lubricant and prevent rapid wear of the PEO layer. Despite the noted disadvantages of the abrasive action of nano-inclusions, the idea of introducing nanoparticles into electrolytes for the synthesis of PEO layers is interesting and promising for further development.

## 4. Conclusions

The PEO layers were synthesized on cast Al–Si alloys. It has been established that PEO layers grow unevenly, and silicon crystals slow their growth. This can be explained by the semiconductor properties of Si and $SiO_2$, as well as their high melting points and heat capacities.

Additionally, it has been established that the silicon content in the cross section of the oxide ceramic layer somewhat increases compared to the AK9 and AK12 alloys in the initial state. This is due to the possibility of introducing silicon into PEO layers from an electrolyte containing $Na_2SiO_3$.

XRD analysis shows that the synthesized PEO layers contain the following phases: aluminum oxides ($\alpha$-$Al_2O_3$ and $\gamma$-$Al_2O_3$), as well as sillimanite ($Al_2O_3 \cdot SiO_2$), mullite ($3Al_2O_3 \cdot 2SiO_2$), and silicon oxide.

The addition of hydrogen peroxide to the electrolyte promotes the intense formation of high-temperature phases, in particular $\alpha$-$Al_2O_3$ (corundum) and $3Al_2O_3 \cdot 2SiO_2$ (mullite).

Furthermore, the abrasion wear resistance of PEO layers on AK9 and AK12 alloys increases by 14–57 times compared to the alloys in their initial state.

The addition of 3–5 g/L of hydrogen peroxide to the basic electrolyte makes it possible to additionally increase the wear resistance of the PEO layers by 30–70%. Most likely, this is due to an increase in the corundum content in the PEO layers and a decrease in the pore size.

**Author Contributions:** Conceptualization, M.S. and I.P.; methodology, V.P., H.V., A.K. and K.Z.; validation, V.P., I.K., V.H., A.K. and K.Z.; formal analysis, I.P., J.P. and H.V.; investigation, V.P., V.H., I.K. and K.Z.; Interpretation of results, M.S., I.P., J.P., H.V., A.K. and H.C.; resources, J.P. and I.P.; data curation, I.P., M.S., V.H. and J.P.; writing—original draft preparation, M.S., V.P., K.Z. and I.K.; discussion, M.S., I.P., I.K., J.P., V.P. and H.V.; writing—review and editing, M.S., V.P., J.P. and H.C.; visualization, H.V., I.K., A.K. and K.Z.; supervision, I.P. and M.S.; project administration, J.P. and H.C.; funding acquisition, J.P. All authors have read and agreed to the published version of the manuscript.

**Funding:** The results of studies described in the article, were obtained within the framework of the budget financing program of the National Academy of Sciences of Ukraine "Support for state-priority scientific research and scientific and technical (experimental) developments for 2022–2023." (UPKIK 6541230), approved by the resolution of the Presidium of the National Academy of Sciences of Ukraine No. 214 dated 13 July 2022 and supplemented by Decree No. 280 dated 21 September 2022. This support is highly appreciated by the authors.

**Institutional Review Board Statement:** Not applicable.

**Informed Consent Statement:** Not applicable.

**Data Availability Statement:** The data presented in this study are available upon request from the corresponding author.

**Conflicts of Interest:** The authors declare no conflict of interest.

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
