# Peer review of "Influence of Plasma Electrolytic Oxidation of Cast Al-Si Alloys on Their Phase Composition and Abrasive Wear Resistance"

_coatings, doi:10.3390/coatings13030637_

Round 1

Reviewer 1 Report

It is very well written paper and the research is valuable and the paper is recommended for publication subject minor corrections:

1. Acronym should be introduced before its use. check this is the case for this paper;

2. For readability and clarity, some of Label for X axis should be more explicit rather than number 1, 2, 3, ...; For example, Fig. 15;

3. Correct typo errors such as Reference: No. 31

Author Response

We would like to thank the Reviewers for the valuable remarks to our manuscript. We have revised the whole paper as recommended in the revision. The changes in the manuscript are written in yellow. You will find our answers and comments (text in red colour) to the Reviewers question bellow.

Reviewer 1

  1. Acronym should be introduced before its use. check this is the case for this paper.

The abbreviation of PEO was added to the abstract.

  1. For readability and clarity, some of Label for X axis should be more explicit rather than number 1, 2, 3, ...; For example, Fig. 15.

Figures were corrected.

  1. Correct typo errors such as Reference: No. 31.

The errors were corrected.

Reviewer 2 Report

The authors reported the structural microhardness and abrasive wear resistance of aluminum casting alloys AlSi AK9 and AK12 silumins

The authors should improve some parts of the article or explain more clearly to the readers.

1.         The abbreviation PEG introduced in the abstract should be explained. This was done in the further part of the manuscript but generally, the first abbreviation should be developed.

2.         The supplier of aluminum alloys should be provided.

3.         The time (length) of cathodic and anodic pulses should be provided. The element composition of aluminum alloys is varied (e.q. 84,1...95,6 mass % Al). The authors should it explain. Is there any certificate supplied by the manufacturer or did the authors determined it?

4.         The numbers in the chemical formula should be in subscripts.

5.         All Tables should be enumerated and should contain their own captions.

6.         More details should be provided about IMPELOM equipment (Figure 1) since it strongly deviates from the standard electrolytic cell (there is a lack of auxiliary and reference electrodes).  All details about symbols used in Figure1b should be explained.

7.         There are repetitions in the sentence: “according to the method according to the following scheme”. The sentence: “X-ray phase analysis of PEO layers was performed using a DRON 3.0M diffractometer (Cu-K radiation)” was introduced fifth line above. All repetitions should be eliminated from the text.

8.         The names of manufacturers of all instruments applied in this work should be provided. A more detailed explanation of the scheme shown in Figure 2 is needed or some appropriate references should be provided.

9.         The title of part 3 “Structure of PEO layers synthesized on AK12 alloy” suggest that PEO layer was synthesized on the AK12 while the results were provided for AK9.Figure 3a does not show distribution of elements while it is SEM image. In the text there is lack of information about that technique. It should be added. The thickness of PEO layer should be determined, and some statistics should be provided. The table located at Figure 3a should be separated and numerated from the Figure 3a. Caption for that Table is required.  I guess that the elemental analysis was determined by the EDS. More details should be provided. Normally, carbon determination by that technique is not possible. The authors should provide some explanation.

10.       The sentence: “An additional factor leading to the formation of a network of microcracks is the rapid cooling of the newly formed aluminum oxide by the electrolyte (Figure. 4).” Is unclear. There is no information about the applied cooling method in the Section 2 (materials and method).

11.       The sentence: “The phase composition of the coatings formed in the electrolyte of 3 g/l KOH + 2 g/l Na2SiO3 and in the same electrolyte with the addition of 3 g/l H2O2 is given in Table 1 and Table 2” is difficult to understand.

12.       Instead of “Hydrogen peroxide contributes to the synthesis” it is better: Hydrogen peroxide introduces to the PEO layer a larger amount of the high-temperature …. More similar typos should be carefully revised by all authors and corrected.

13.       Figure 4a – see comment 9.

14.       Description of peaks in XRD spectra (Figure 5 and 6) is mandatory.

15.       What is the difference between Figure 3a and Figure 7 ? Figure 3a and Figure 7 show the PEO layer deposited from the same electrolyte 3 g/l KOH+2 g/l Na2SiO3 while the EDS spectrum in Figure 7 shows a very low carbon peak, significantly lower than Si. On contrary, the concentration of C is greater than Si (Figure3). How to explain it?

A similar remark is related to Figure 8.

16.       The sentence:” Spectral analysis of the area of the PEO layer on the AK9 alloy in 3 g/l KOH+2 g/l Na2SiO3+3g/l KOH revealed a pattern similar to that in the electrolyte 3 g/l KOH+2 g/l Na2SiO3 (Figure. 7) .” is unclear.

17.       Detailed and improved discussion about results shown in Figures 7 and 8 should be provided.

18.       The sentence “Structure of PEO layers synthesized on AK12 alloy. The PEO layer on the AK12 alloy grows deeper into the aluminum matrix more unevenly than on the AK9 alloy (Figure. 9).” is unclear to me. How the PEO layer deposited by the electrolytic method can grow (incorporate) into the substrate? It seems that the thickness of PEO layer on AK12 (Fig.9) is significantly thinner on AK9 due to the higher silicon content in Al substrate. For me, it is not clear why the thickness of PEO layer (Figure9) is lower than for PEO on AK9 since the process was galvanostatically controlled (the same cathodic current density was applied). The authors recorded the applied potential ?

19.       Figure 9 – see comment 9.

20.       The expected electrochemical reactions that occur during electrolysis should be provided. The role of H2O2 should be explained in this context.

21.       Figure 11 – see comment 9.

22.       Figure 13, 14 – see comment 14.

23.       The substrates: AMG6 and D16 (Figure 15) were not specified in Section 2.

24.       Some hypotheses about the influence of H2O2 concentration of wear (Figure 16) should be provided.

25.       The sentence:” Thus, the wear resistance of the PEO layer on the AK9 alloy, synthesized in the base electrolyte with the addition of hydrogen peroxide, increased 100 times compared to the AK9 alloy.” is unclear.

26.       The role of H2O2 in increasing wear resistance (Fig.19) is not properly explained.

Author Response

We would like to thank the Reviewers for the valuable remarks to our manuscript. We have revised the whole paper as recommended in the revision. The changes in the manuscript are written in yellow. You will find our answers and comments (text in red colour) to the Reviewers question bellow.

Reviewer 2

The authors should improve some parts of the article or explain more clearly to the readers.

  1. The abbreviation PEO introduced in the abstract should be explained. This was done in the further part of the manuscript but generally, the first abbreviation should be developed.

The abbreviation PEO was added in the abstract.

  1. The supplier of aluminum alloys should be provided.

The supplier was KYEVTSVETMET LLC (Kyiv, Ukraine).

  1. The time (length) of cathodic and anodic pulses should be provided. The element composition of aluminum alloys is varied (e.q. 84,1...95,6 mass % Al). The authors should it explain. Is there any certificate supplied by the manufacturer or did the authors determined it?

The time of anodic and cathodic pulses was 6 msec. The IMPELOM equipment was developed at our institute.

  1. The numbers in the chemical formula should be in subscripts.

The numbers in the formulas were added to subscripts.

  1. All Tables should be enumerated and should contain their own captions.

Tables were numbered.

  1. More details should be provided about IMPELOM equipment (Figure 1) since it strongly deviates from the standard electrolytic cell (there is a lack of auxiliary and reference electrodes).  All details about symbols used in Figure1b should be explained.

Fig. 1 was deleted because the scheme of IMPELOM equipment is a standard one for plasma electrolytic oxidation.

  1. There are repetitions in the sentence: “according to the method according to the following scheme”. The sentence: “X-ray phase analysis of PEO layers was performed using a DRON 3.0M diffractometer (Cu-Ka radiation)” was introduced fifth line above. All repetitions should be eliminated from the text.a There are repetitions in the sentence: “according to the method according to the following scheme”. The sentence: “X-ray phase analysis of PEO layers was performed using a DRON 3.0M diffractometer (Cu-Kα).

The repetitions were excluded.

  1. The names of manufacturers of all instruments applied in this work should be provided. A more detailed explanation of the scheme shown in Figure 2 is needed or some appropriate references should be provided.

The methods of investigations of the abrasive wear resistance are researched in Ukraine and at our institute (Karpenko Physico-Mechanical Institute of the National Academy of Sciences of Ukraine, Lviv, Ukraine). The obtained results were published in a large number of articles.

  1. The title of part 3 “Structure of PEO layers synthesized on AK12 alloy” suggest that PEO layer was synthesized on the AK12 while the results were provided for AK9.Figure 3a does not show distribution of elements while it is SEM image. In the text there is lack of information about that technique. It should be added. The thickness of PEO layer should be determined, and some statistics should be provided. The table located at Figure 3a should be separated and numerated from the Figure 3a. Caption for that Table is required.  I guess that the elemental analysis was determined by the EDS. More details should be provided. Normally, carbon determination by that technique is not possible. The authors should provide some explanation.

We have added a description of the method of determining distribution of elements. We did not determine the thickness of the PEO layers. Increasing silicon content reduces the thickness of the PEO layers, we investigated this in our previous work [Kovalchuk I.V., Yurkevych R.M., Posuvailo V.M. Crystal structure of oxide ceramic coatings obtained on alloys with a high silicon content. Physicochemical Mechanics of Materials. 2022, Vol. 58. P. 50-57]. Also, as you noted, the anode potentials of PEO coatings are decreased.

  1. The sentence: “An additional factor leading to the formation of a network of microcracks is the rapid cooling of the newly formed aluminum oxide by the electrolyte (Figure. 4).” Is unclear. There is no information about the applied cooling method in the Section 2 (materials and method).

In the Section 2 (Materials and Method) the information on the cooling method does not provide. We synthesized the coating by the plasma electrolytic oxidation method. A high anodic voltage is applied to the valve metal (Al, Mg, Ti) immersed in the electrolyte. When the voltage exceeds 100 V, a breakdown of the primary oxide film is occured on the valve metals. A plasma channel with high temperature is formed. The in high-temperature oxides are formed near plasma channels. The oxides melted by the plasma discharge are cooled by the electrolyte after the attenuation of discharge channel. This can be the reason for the formation of microcracks in the coating.

  1. The sentence: “The phase composition of the coatings formed in the electrolyte of 3 g/l KOH + 2 g/l Na2SiO3 and in the same electrolyte with the addition of 3 g/l H2O2 is given in Table 1 and Table 2” is difficult to understand.

The sentence was corrected.

  1. Instead of “Hydrogen peroxide contributes to the synthesis” it is better: Hydrogen peroxide introduces to the PEO layer a larger amount of the high-temperature. More similar typos should be carefully revised by all authors and corrected.

The effect of hydrogen peroxide is expanded in the text of the article in section 3.3.

  1. Figure 4a – see comment 9.

The description of the equipments was added to the definition of elements, and the methods and materials section was added. We have added a description of the method of determining distribution of elements. We did not determine the thickness of the PEO layers. Increasing silicon content reduces the thickness of the PEO layers, we investigated this in our previous work [Kovalchuk I.V., Yurkevych R.M., Posuvailo V.M. Crystal structure of oxide ceramic coatings obtained on alloys with a high silicon content. Physicochemical Mechanics of Materials. 2022, Vol. 58. P. 50-57]. Also, as you noted, the anode potentials of PEO coatings are decreased.

  1. Description of peaks in XRD spectra (Figure 5 and 6) is mandatory.

All XRD patterns were refined by Rietveld method using FullProf software.

  1. What is the difference between Figure 3a and Figure 7 ? Figure 3a and Figure 7 show the PEO layer deposited from the same electrolyte 3 g/l KOH+2 g/l Na2SiO3 while the EDS spectrum in Figure 7 shows a very low carbon peak, significantly lower than Si. On contrary, the concentration of C is greater than Si (Figure3). How to explain it?

The distribution of elements was refined.

16.The sentence:” Spectral analysis of the area of the PEO layer on the AK9 alloy in 3 g/l KOH+2 g/l Na2SiO3+3g/l KOH revealed a pattern similar to that in the electrolyte 3 g/l KOH+2 g/l Na2SiO3 (Figure. 7) .” is unclear.

We corrected the mistake in the description of the electrolyte.

17.Detailed and improved discussion about results shown in Figures 7 and 8 should be provided.

The results were added.

18.The sentence “Structure of PEO layers synthesized on AK12 alloy. The PEO layer on the AK12 alloy grows deeper into the aluminum matrix more unevenly than on the AK9 alloy (Figure. 9).” is unclear to me. How the PEO layer deposited by the electrolytic method can grow (incorporate) into the substrate? It seems that the thickness of PEO layer on AK12 (Fig.9) is significantly thinner on AK9 due to the higher silicon content in Al substrate. For me, it is not clear why the thickness of PEO layer (Figure9) is lower than for PEO on AK9 since the process was galvanostatically controlled (the same cathodic current density was applied). The authors recorded the applied potential ?

The explanation is in the text of the article. We control the process by current density. As the amount of silicon in the alloys increases, the potential decreases [Kovalchuk I.V., Yurkevych R.M., Posuvailo V.M. Crystal structure of oxide ceramic coatings obtained on alloys with a high silicon content. Physicochemical Mechanics of Materials. 2022, Vol. 58. P. 50-57].

19.Figure 9 – see comment 9.

We have added a description of the method of determining the distribution of elements. We did not determine the thickness of the peo layers separately. Increasing the silicon content reduces the thickness of the PEO layers, we investigated this in our previous work [Kovalchuk I.V., Yurkevych R.M., Posuvailo V.M. Crystal structure of oxide ceramic coatings obtained on alloys with a high silicon content. Physicochemical Mechanics of Materials. 2022, Vol. 58. P. 50-57]. Also, as you correctly noted, the anode potentials decrease.

20.The expected electrochemical reactions that occur during electrolysis should be provided. The role of H2O2 should be explained in this context.

The effect of H2O2 is analyzed in detail in the works:

Posuvailo V. M.; Kulyk V. V.; Duriagina Z. A.; Koval’chuck I. V.; Student M. M.; Vasyliv B. D. The effect of electrolyte composition on the plasma electrolyte oxidation and phase composition of oxide ceramic coatings formed on 2024 aluminium alloy. Archives of Mat. Scie. and Eng. 2020. 105. 49–55 DOI:10.5604/01.3001.0014.5761

Нutsaylyuk V.; Student M.; Posuvailo V.; Student O.; Maruchak P. Zakiev V. The role of hydrogen in the formation of oxide-ceramic layers on aluminum alloys during their plasma-electrolytic oxidation. J. Mater. Resear. Techn. 2021, 14 1682-1696. https://doi.org/10.1016/j.jmrt.2021.07.082.

21.Figure 11 – see comment 9.

Figure 11 was corrected.

22.Figure 13, 14 – see comment 14.

Figure 13 and 14 were corrected:

Posuvailo V. M.; Kulyk V. V.; Duriagina Z. A.; Koval’chuck I. V.; Student M. M.; Vasyliv B. D. The effect of electrolyte composition on the plasma electrolyte oxidation and phase composition of oxide ceramic coatings formed on 2024 aluminium alloy. Archives of Mat. Scie. and Eng. 2020. 105. 49–55 DOI:10.5604/01.3001.0014.5761

Student М. М.; Dovhunyk V. M.; Posuvailo V. M.; Koval’chuk I. V.; Hvozdets’kyi V. M. Friction behavior of iron-carbon alloys in couples with plasma-electrolytic oxide-ceramic layers synthesized on D16T alloy. Mater. Sci. 2017, 53. 359–367 https://doi.org/10.1007/s11003-017-0083-x

23.The substrates: AMG6 and D16 (Figure 15) were not specified in Section 2.

It is indicated in the text of the article.

24.Some hypotheses about the influence of H2O2 concentration of wear (Figure 16) should be provided.

3-5 g/l H2O2 increases the corundum content in PEO layer:

Posuvailo V. M.; Kulyk V. V.; Duriagina Z. A.; Koval’chuck I. V.; Student M. M.; Vasyliv B. D. The effect of electrolyte composition on the plasma electrolyte oxidation and phase composition of oxide ceramic coatings formed on 2024 aluminium alloy. Archives of Mat. Scie. and Eng. 2020. 105. 49–55 DOI:10.5604/01.3001.0014.5761

  1. The sentence:” Thus, the wear resistance of the PEO layer on the AK9 alloy, synthesized in the base electrolyte with the addition of hydrogen peroxide, increased 100 times compared to the AK9 alloy.” is unclear.

The sentence was corrected.

26.The role of H2O2 in increasing wear resistance (Fig.19) is not properly explained.

3-5 g/l H2O2 increases the corundum content in PEO layer:

  1. Posuvailo V. M.; Kulyk V.; Duriagina Z. A.; Koval’chuck I. V.; Student M. M.; Vasyliv B. D. The effect of electrolyte composition on the plasma electrolyte oxidation and phase composition of oxide ceramic coatings formed on 2024 aluminium alloy. Archives of Mat. Scie. and Eng. 2020. 105. 49–55 DOI:10.5604/01.3001.0014.5761
  2. Hutsaylyuk V.; Student M.; Posuvailo V.; Student O.; Maruchak P. Zakiev V. The role of hydrogen in the formation of oxide-ceramic layers on aluminum alloys during their plasma-electrolytic oxidation. J. Mater. Resear. Techn. 2021, 14 1682-1696. https://doi.org/10.1016/j.jmrt.2021.07.082

Reviewer 3 Report

The authors reported the structural microhardness and abrasive wear resistance of aluminum casting alloys Al-Si AK9 and AK12 silumins. The manuscript is well-written and well-organized as well as the English language is fine and should be accepted for publication after minor revision and according to the following suggestions: -

 1- The Abstract is concise and well-written but the subscript numbers for Al2O3 and SiO2 chemical compounds should be modified. Also, “up to 900… 1000 HV” and “14 ... 57 times” should be modified.

2- Materials and methods, the bottom of page 2, this part of the sentence “(84,1...95,6 mass % Al; 7,5...10 mass % Si; 0,5…2 mass % Cu; 0,5…0, mass % Ti; 0,2...0,8 mass % Mg; 0,1...0,4 % mass Mn)” should be modified.

3- Materials and methods, page 3, the working conditions for the use of DRON-3.0 X-ray diffractometer using CuKa radiation, EVO 40 XVP electron microscope in characteristic BSD radiation, and PMT-3 microhardness tester should be mentioned.

4- Results and discussion, the last paragraph of page 4 as well as in Table 1 (page 5) and Table 3 (page 9), and throughout other paragraphs of the text of the manuscript; the symbols for chemical compounds; Na2SiO3, Al2O3, SiO2, etc. should be written in the correct format. These symbols are written in the correct way in the figures’ captions and also in Table 2 and Table 4.

5- Conclusions, this part is short and should be modified with more findings.

Author Response

We would like to thank the Reviewers for the valuable remarks to our manuscript. We have revised the whole paper as recommended in the revision. The changes in the manuscript are written in yellow. You will find our answers and comments (text in red colour) to the Reviewers question bellow.

Reviewer 3

The authors reported the structural microhardness and abrasive wear resistance of aluminium casting alloys Al-Si AK9 and AK12 silumins. The manuscript is well-written and well-organized as well as the English language is fine and should be accepted for publication after minor revision and according to the following suggestions:

  • The Abstract is concise and well-written but the subscript numbers for Al2O3 and SiO2 chemical compounds should be modified. Also, “up to 900… 1000 HV” and “14 ... 57 times” should be modified.

The subscript numbers of chemical compounds were corrected. The microhardness and wear resistance were corrected.

  • Materials and methods, the bottom of page 2, this part of the sentence “(84,1...95,6 mass % Al; 7,5...10 mass % Si; 0,5…2 mass % Cu; 0,5…0, mass % Ti; 0,2...0,8 mass % Mg; 0,1...0,4 % mass Mn)” should be modified.

Section Materials and Methods was corrected.

  • Materials and methods, page 3, the working conditions for the use of DRON-3.0 X-ray diffractometer using CuKa radiation, EVO 40 XVP electron microscope in characteristic BSD radiation, and PMT-3 microhardness tester should be mentioned.

The working conditions for the use of DRON-3.0 X-ray diffractometer using CuKa radiation, EVO 40 XVP electron microscope in characteristic BSD radiation, and PMT-3 microhardness tester were added to the section Materials and methods.

  • Results and discussion, the last paragraph of page 4 as well as in Table 1 (page 5) and Table 3 (page 9), and throughout other paragraphs of the text of the manuscript; the symbols for chemical compounds; Na2SiO3, Al2O3, SiO2, etc. should be written in the correct format. These symbols are written in the correct way in the figures’ captions and also in Table 2 and Table 4.

These symbols of chemical compounds were corrected.

5.Conclusions, this part is short and should be modified with more findings.

The conclusions were modified.

Reviewer 4 Report

REPORTS ON: coatings-2165386

The proposed manuscript is reasonably well-organized and reasonably discussed. It has a great number of experimentations and interesting results and conclusions. Based on the results and contribution, the manuscript seems to DESERVES its final publication after a MAJOR REVISION, as following indicated

1.                    Into the Abstract, a simple present tense should be used.

2.                    When AK9 and AK12 are mentioned, including into the proposed title, it is very confusing. This mainly when future reader has not familiarization with nomenclature. Based on this, it is suggested that elucidation be provided. Also, it is hardly suggested that a commonly terminology (industrial or academic) be used. For instance, it is only possible to identify that AK9 and AK12 as Al-Si based alloy when reading the material and method section. This should be elucidate at initial page of the article, including the proposed title or abstract.

3.                    Throughout section 2, the reproducibility is absent or rather/poorly described. At least duplicate should be carried out.

4.                    All values and dimension in section 2 should be accompanied with their error ranges.

5.                    What are reproducibility for those results shown in Figs. 3, 4, 5, 6 and 9 and 11? This should obligatory be clarified.

6.                    JCPDS or similar file numbers should be elucidated in Figs. 5 and 6.

7.                    The section 3 has some subdivisions, which are not numbered.

8.                    In Figs. 5 and 6,and 13 and 14, Rietveld treatments are provided. However, this is not detailed in caption or discussion text correspondent.

9.                    Figs 15 to 19 should be revised and reorganized. It seems that some of those figures can be merged in unique panel. Besides, error ranges should be included.

10.                 Since Fig. 15 depicts hardness results, it is at least expected that into section 2 more detail be included.

11.                 Fig 15 is poorly discussed and correlated with both operational parameters experimented and resulting microstructural coatings.

12.                 Finally, but not less important, from those 31 cited references, only 10 dating between 2019 and 2023. It is also hardly suggested that Authors taken attention to include more recently published into COATINGS (MDPI) Journal.

_  _ _ _ _

Author Response

We would like to thank the Reviewers for the valuable remarks to our manuscript. We have revised the whole paper as recommended in the revision. The changes in the manuscript are written in yellow. You will find our answers and comments (text in red colour) to the Reviewers question bellow.

Reviewer 4

  1. Into the Abstract, a simple present tense should be used.

We used the Past Simple and Present Simple Tenses.

  1. When AK9 and AK12 are mentioned, including into the proposed title, it is very confusing. This mainly when future reader has not familiarization with nomenclature. Based on this, it is suggested that elucidation be provided. Also, it is hardly suggested that a commonly terminology (industrial or academic) be used. For instance, it is only possible to identify that AK9 and AK12 as Al-Si based alloy when reading the material and method section. This should be elucidate at initial page of the article, including the proposed title or abstract.

The article title was corrected. The alloys and their analogues were added to section Materials and Methods.

3.Throughout section 2, the reproducibility is absent or rather/poorly described. At least duplicate should be carried out.

We performed the statistical processing of the results of measuring microhardness and abrasive wear resistance of PEO coatings. The error bars were added in the figures. The initial alloys AK9 and AK12 have a high content of silicon and its uneven distribution in the alloys. As a result, the distribution of silicon across the cross sections of the PEO layer can vary greatly on small areas. Although the distribution of silicon is almost the same on a large cross-sectional areas.

4.All values and dimension in section 2 should be accompanied with their error ranges.

The range of errors were added in the figures. The accuracy of determining phase composition by the Rietveld method is 2–3 wt.%.

  1. What are reproducibility for those results shown in Figs. 3, 4, 5, 6 and 9 and 11? This should obligatory be clarified.

The initial alloys AK9 and AK12 have a high content of silicon and its uneven distribution in the alloys. As a result, the distribution of silicon across the cross sections of the PEO layer can vary greatly on small areas. Although the distribution of silicon is almost the same on a large cross-sectional areas. It also leads to a difference in the phase composition. The accuracy of determining phase composition by the Rietveld method is 2–3 wt.%.

  1. JCPDS or similar file numbers should be elucidated in Figs. 5 and 6.

Known structures of compounds were used to calculate the phase content:

  1. Schneider H.; Fischer R. X.; Voll D. Mullite with lattice constants a > b. J. Am. Ceram. Soc. 1993. 76. 1879–1881. https://doi.org/10.1111/j.1151-2916.1993.tb06666.x
  2. Schneider H.; Schreuer J.; Hildman B. Structure and properties of mullite – A review. J. Europ. Ceram. Soc. 2008, 28. 329–344. http://dx.doi.org/10.1016/j.jeurceramsoc.2007.03.017
  3. Burnharn C. W. Refinement of crystal structure of sillimanite. Z. Krist. 1963. – 118. 127–148 1524/zkri.1963.118.1-2.127
  4. section 3 has some subdivisions, which are not numbered.

The subdivisions were numbered in Section 3.

  1. In Figs. 5 and 6,and 13 and 14, Rietveld treatments are provided. However, this is not detailed in caption or discussion text correspondent.

The phase composition of the PEO layers calculated by the Rietveld method is given in the tables.

  1. Figs 15 to 19 should be revised and reorganized. It seems that some of those figures can be merged in unique panel. Besides, error ranges should be included.

The figures were reorganized.

  1. Since Fig. 15 depicts hardness results, it is at least expected that into section 2 more detail be included.

An analysis of the microhardness of PEO layers obtained on various alloys was added to Chapter 3.

  1. Fig 15 is poorly discussed and correlated with both operational parameters experimented and resulting microstructural coatings.

An analysis of the microhardness of oxide ceramic coatings was added to the text of the article.

  1. Finally, but not less important, from those 31 cited references, only 10 dating between 2019 and 2023. It is also hardly suggested that Authors taken attention to include more recently published into COATINGS (MDPI) Journal.

The new articles published in Coatings MDPI were included in the References.

Reviewer 5 Report

The third version of the paper looks quite well, but the manuscript needs some additional corrections. There are many points to correct and/or improve in the manuscript, as indicated in the PDF file with color highlights. Please look at the following: lines 30, 61, 93-96, 102, 116, 126, 132, 133, 142, 145, 149, below 178, 198, below 221, below 250, 263, 295, 321, 422, 430; also Tables 1, 2, 3, 4 -- and introduce changes//corrections.

     Here are some examples, for instance: unify your unit "wt%", instead of >>wt. %<<, >>wt %<<, or >>Wt. %<< (per cent is not a unit here). Line 106: if the current intensity is denoted by "I", so the current density used to be denoted by "i" (not >>j<<). Line 116: note that it is "Vickers hardness test" (not >>Vicker's...<<).

Author Response

Manuscript ID: coatings-2165386

Influence of plasma electrolytic oxidation of cast Al-Si alloys on their phase composition and abrasive wear resistance

To the editorial office of the Scientific Journal of “Coatings”

We are very grateful to the distinguished editor and reviewers for the professional analysis of our work presented for publication in the journal "Coatings". We were carefully studied all remarks, comments and wishes, and tried to take them into account as much as possible, making the necessary additions to the text of the article. In addition, following the recommendation of the reviewers regarding the quality of the translation into English and understanding its importance for perceive written, the text of our article was edited by a native English speaker. All additions to the article, which arose as a reaction to the comments of the reviewers, we made directly to the text. They are highlighted in gray. What should be removed from the text of the article is highlighted in red.

Thanks again for your understanding and patience while reviewing our article.

best wishes

The units of measurement given in the article were unified.

Vicker’s hardness tests in the article has been corrected.

Reviewer 6 Report

Unfortunately, the article has serious flaws and can not be published unless it is deeply reworked.

1) One of the main conclusions is that "the silicon content decreases by approximately two times compared to AK9 and AK12 alloys in the initial state". Let’s check that based on the data from Figure 2b, for example.

The initial allow is AK9 that has 9wt% of Si, i.e. Si to Al ratio = 9:91 = 1:10. After the treatment we have Si to Al = 5.14 : 41.27 = 1:8. It means that Si content increased (!) compared to the initial material.

2) EDX numbers should be double checked. For instance, in Fig. 7b we have Si:Al = 4.29 : 42.33 = 0.1 (weight) and 1.58 : 31.77 = 0.05 (atomic). That is obviously incorrect since atomic weights of Si and Al are almost the same.

3) There is a problem in oxygen content. For instance, let’s analyze Fig. 4b. If we suppose a fully oxidized material, we would have 30.8*0.5*(Al2O3) + 3.73*(SiO2) + 0.56*0.5*(K2O) = 54 at% of oxygen. From the data, we have 65 at% - that is a big discrepancy.

4) I don't understand the description of AKx alloys. For instance, AK12 can have up to 95.6% of Al. If it has that much Al, how it can have 10 or more % of Si?

5) AK alloys can have varied concentration of elements. So, if you want to compare the composition of the film and composition of the substrate, you have to measure the both regions.

6) It is strange to present EDX tables as parts of the figures.

7) It is strange that the initial submission has differently colored and highlighted text. Sometimes, there is an obvious mess (e.g. lines 130-133).

8) Rietveld analysis can not be done on thin films. This is a technique for powder samples, where crystal orientations are fully random. That can be one of the reasons why the residual pattern has peaks with very high intensities.

9) XRD phase analysis contradicts EDX data. For instance, from Table 3 you have approximately 6.4+45.37+63.69*2/3 = 94.23 wt% of Al2O3 and 4.17 + 63.69/3 = 25.4 wt% of SiO2. (By the way, why the sum in the table is not 100%?) That means Si to Al ~ 25.4/60 : 94.23/102/2 = 0.23. From EDX we have only 3.5/30.87 = 0.11.

10) I have a strong doubt that the resolution of 0.05 deg is sufficient to analyze such complex systems.

11) XRD data are badly presented. All observed peaks should be indexed (hkl) and referred to a certain phase. Their positions should be shown. Probably, that can be done in a separate table. Reference powder spectra must include peak intensities. At least, the reference spectra must be marked!

12) Lattice constant are given with very high accuracy. On the other hand, they differ greatly from sample to sample. For instance, c for SiO2 can be 5.505, 5.7, 5.734, while for the normal crystal it is 5.405. It seems that the real accuracy is much lower.

13) Suitable references are required for crystallographic parameters of all phases observed in XRD.

14) In the tables, the lattice constants should be small letters and you must include unit of measurement.

15) Lines 272-273. The explanation can not be correct. First, the penetration depth of Cu Kalpha X-rays in Al2O3 is just few tens of um, that is much smaller than the coating thickness. Second, the inclusion of Si, at which the growth stops, is Si, not SiO2 (form EDX maps).

15) How did you measure the thickness of such inhomogeneous layers? Also, in line 247: “increase in the number of microcracks and pores”, how did you evaluate the number of cracks and pores?

16) Figure 13. The thickness is identical within the experimental error. There is nothing to discuss.

Author Response

Manuscript ID: coatings-2165386

Influence of plasma electrolytic oxidation of cast Al-Si alloys on their phase composition and abrasive wear resistance

To the editorial office of the Scientific Journal of “Coatings”

We are very grateful to the distinguished editor and reviewers for the professional analysis of our work presented for publication in the journal "Coatings". We were carefully studied all remarks, comments and wishes, and tried to take them into account as much as possible, making the necessary additions to the text of the article. In addition, following the recommendation of the reviewers regarding the quality of the translation into English and understanding its importance for perceive written, the text of our article was edited by a native English speaker. All additions to the article, which arose as a reaction to the comments of the reviewers, we made directly to the text. They are highlighted in gray. What should be removed from the text of the article is highlighted in red.

Thanks again for your understanding and patience while reviewing our article.

best wishes

Below are our responses to the remarks and comments of distinguished reviewers:

1) One of the main conclusions is that "the silicon content decreases by approximately two times compared to AK9 and AK12 alloys in the initial state". Let’s check that based on the data from Figure 2b, for example.

The initial allow is AK9 that has 9wt% of Si, i.e. Si to Al ratio = 9:91 = 1:10. After the treatment we have Si to Al = 5.14 : 41.27 = 1:8. It means that Si content increased (!) compared to the initial material.

In general, the authors agree with the reviewer's considerations. To check their validity, the following additional experiments were carried out. From the analysis of X-ray patterns obtained for these alloys from an area of 4 ... 5 mm2 (X-ray beam diameter was 2.5 mm), the integral chemical composition of both analyzed aluminum alloys (AK9 and AK12) was determined and presented in the article. It was confirmed that in both alloys, the silicon crystals are unevenly distributed (hence, there is a significant segregation in the silicon content in the substrates). Therefore, it is clear that the results of spectral analysis of small (up to 0.01 mm2) areas (tabular data of X-ray microspectral analysis in fig. 3 in the edited text of the article) of these alloys could differ significantly in silicon content relative to its integral content. This depended on whether large silicon crystals were encountered in the analyzed regions. The spectral analysis of the PEO layer and the substrate under it confirmed an increase in the silicon content in the PEO layer (tabular data of X-ray microspectral analysis in fig. 3 in the edited text of the article). It was believed that despite the decrease in the silicon content in the PEO layer (due to the formation of volatile SIO and SIH phases), it simultaneously increased (due to the incorporation of silicon from the Na2SiO3 compound present in the electrolyte). The authors are grateful to the reviewer for his remark. The work on taking it into account made it possible to better understand and describe the features of the processes that can occur during the synthesis of PEO layers.

2) EDX numbers should be double checked. For instance, in Fig. 7b we have Si:Al = 4.29 : 42.33 = 0.1 (weight) and 1.58 : 31.77 = 0.05 (atomic). That is obviously incorrect since atomic weights of Si and Al are almost the same.

The authors agreed with the comment of the reviewer. To reduce the discrepancy between the content of the main components in the PEO layer (Si and Al), the content of these components was additionally analyzed (according to the results of the analysis of areas of a larger area) and the consistency of their compositions in mass and atomic percent was achieved.

3) There is a problem in oxygen content. For instance, let’s analyze Fig. 4b. If we suppose a fully oxidized material, we would have 30.8*0.5*(Al2O3) + 3.73*(SiO2) + 0.56*0.5*(K2O) = 54 at% of oxygen. From the data, we have 65 at% - that is a big discrepancy.

The authors agreed with the comment of the reviewer. To reduce the discrepancy between the content of the main components in the PEO layer (Si and Al), we additionally analyzed the content of these components by analyzing large areas, and obtained a satisfactory correspondence between their ratios in terms of mass and atomic percentages (tabular data of X-ray microspectral analysis in fig. 3 in the edited text of the article).

4) I don't understand the description of AKx alloys. For instance, AK12 can have up to 95.6% of Al. If it has that much Al, how it can have 10 or more % of Si?

Thanks to the reviewer for pointing out the typo. To confirm the integral content of elements in both analyzed alloys AK9 and AK 12, their content was determined again by analyzing X-ray diffraction patterns obtained from an area of 4–5 mm2 (the diameter of the X-ray beam was 2.5 mm). Since, in this case, X-rays penetrate into the depth of the PEO layers, this gave reason to believe that a certain content of elements in the composition of AK9 and AK 12 alloys is integral. This content is given in the text of the article. (Oxide ceramic layers were synthesized on two cast Al-Si alloys of the following composition, wt %:, namely AK9 (88.8 Al; 9.9 Si; 0.5 Cu; 0. 5 Ti; 0.2 Mg; 0.1 Mn) and AK12  (85.2 Al; 13.5 Si; 0.7 Fe; 0.6 Cu.)

5) AK alloys can have varied concentration of elements. So, if you want to compare the composition of the film and composition of the substrate, you have to measure the both regions.

The authors agree with the reviewer's comment. To compare the composition of the PEO layers and the corresponding substrates, we performed an X-ray microanalysis of the equivalent areas of the PEO layers and substrates. The results obtained are presented in the text of the article (tabular data of X-ray microspectral analysis in fig. 3 in the edited text of the article).

6) It is strange to present EDX tables as parts of the figures.

This remark of the reviewer was considered recommendatory. In the technical literature, there are different options for presenting the results of X-ray microanalysis. The main requirement is that they must be understandable. However, taking into account the wishes of the reviewer, the results of X-ray microanalysis were presented in the text of the article as part of a drawing.

7) It is strange that the initial submission has differently colored and highlighted text. Sometimes, there is an obvious mess (e.g. lines 130-133).

Reviewer's comment taken into account

8) Rietveld analysis can not be done on thin films. This is a technique for powder samples, where crystal orientations are fully random. That can be one of the reasons why the residual pattern has peaks with very high intensities.

In response to the reviewer's remarks, the authors make the following comment. The synthesized PEO layers are not a film, but rather thick (about 200 µm) layers. They have high adhesion to the substrate and cannot be torn off and ground into powder. The appearance of high peaks in the difference X-ray diffraction patterns was due to the defective structure of the silimanite and mullite compounds. Unfortunately, it was not possible to accurately determine the degree of occupation of oxygen sites in these compounds.

9) XRD phase analysis contradicts EDX data. For instance, from Table 3 you have approximately 6.4+45.37+63.69*2/3 = 94.23 wt% of Al2O3 and 4.17 + 63.69/3 = 25.4 wt% of SiO2. (By the way, why the sum in the table is not 100%?) That means Si to Al ~ 25.4/60 : 94.23/102/2 = 0.23. From EDX we have only 3.5/30.87 = 0.11.

The authors agreed with the reviewer's remark about the discrepancy between the XRD phase and the EDX analysis of the synthesized PEO layers. This is due to the fact that EDX microanalysis was carried out on small areas of PEO layers (up to 0.02 mm2), and XRD phase analysis was carried out from an area of 4–5 mm2. Additional studies using the EDX method showed that due to the significant segregation of large silicon crystals in cast alloys, silicon was also distributed very unevenly in the PEO layers (Fig. 9 for AK9 and Fig. 14 for AK12 alloys). Therefore, analysis of the content of elements in a small area gave a significant scatter of data (depending on the location of the analysis). An additional analysis of the EDX over a larger area (up to 0.15 mm2) made it possible to practically level the discrepancy in the assessment of the silicon content by both methods. The article replaces the photographs and presents the corresponding EDX analysis data from larger areas. Due to this, a satisfactory agreement between the data of both variants of the analysis of the alloys under study was achieved.

To the comment of the reviewer about the lack of equality of the sum of the constituent layers of PEO 100% in Table. 3, this is an unfortunate error, and the authors are grateful to the reviewer for the fact that, thanks to his attentiveness, it was possible to eliminate it.

10) I have a strong doubt that the resolution of 0.05 deg is sufficient to analyze such complex systems.

The authors agree with the reviewer's remark that it is better to use precision X-ray analysis for the analyzed structures, in particular, using synchrotron radiation. However, today (due to lack of time, unpredictable power outages and Internet communications in Ukraine, frequent forced stays in bomb shelters), it is not possible to clarify the phase composition of PEO layers using such an analysis. However, the phase composition determined and given in the text of the article gives an idea of the possible mechanism for the formation of PEO layers on cast aluminum alloys and, most importantly, shows a possible way to increase their wear resistance.

11) XRD data are badly presented. All observed peaks should be indexed (hkl) and referred to a certain phase. Their positions should be shown. Probably, that can be done in a separate table. Reference powder spectra must include peak intensities. At least, the reference spectra must be marked!

In response to the reviewer's remarks, the authors make the following comment. Many reflections of the detected phases were superimposed on the X-ray patterns (silimanite and mulite are commonly referred to as isostructural phases). To avoid cluttering up the diffractograms with additional inscriptions, the authors considered it not expedient to designate the reflections of each phase. If the reviewer insists on the need for such information in the context of the goals and objectives pursued by this article (increasing the hardness and wear resistance of the surface layers of cast aluminum alloys), then the authors are ready to supplement the article with separate tables with a breakdown of all the identified reflections.

12) Lattice constant are given with very high accuracy. On the other hand, they differ greatly from sample to sample. For instance, c for SiO2 can be 5.505, 5.7, 5.734, while for the normal crystal it is 5.405. It seems that the real accuracy is much lower.

The comments of the reviewer are correct and taken into account. Based on the X-ray diffraction patterns presented in the article, the accuracy of estimating the parameters of the crystal lattice was reduced.

In addition, the lattice parameters of both elements (Al and Si) in their composition and their content (in wt %) were determined by X-ray diffraction study of the analyzed cast alloys (AK9 and AK12).

Typical XRD patterns obtained for cast aluminum alloy AK9 used as a substrate for the synthesis of PEO layers.

The lattice parameters of Al and Si, determined from X-ray diffraction patterns of the AK9 alloy used as a substrate for the synthesis of PEO layers, and the content of these elements (in wt %) are given in table.

Element

a, Å

b, Å

c, Å

PG

wt %

Al

4.0446(5)

4.0446(5)

4.0446(5)

F m -3 m

89

Si

5.4262(5)

5.4262(5)

5.4262(5)

F d -3 m

11

The lattice parameters of Al and Si and their contents (wt %) were determined similarly for another investigated casting alloy AK12.The results obtained are shown in table.

Element

a, Å

b, Å

c, Å

PG

wt %

Al

4.0453(3)

4.0453(3)

4.0453(3)

F m -3 m

86.5

Si

5.4252(1)

5.4252(1)

5.4252(1)

F d -3 m

13.5

The lattice parameters of Si and Al, taken from available crystallographic databases, are 5.4309 Å and 4.0406 Å, respectively. The lattice parameters for both elements, obtained as a result of our X-ray diffraction studies of both casting alloys, are in fairly good agreement with these data.

13) Suitable references are required for crystallographic parameters of all phases observed in XRD.

The crystallographic parameters of all phases revealed in the X-ray diffraction patterns were taken from available literature sources (links to some of them are given below).

  1. For a-Al2O3 (corundum)

Shirai T., Watanabe H., Fuji M., Takahashi M. Structural properties and surface characteristics on aluminum oxide powders. Nagoya Institute of Technology Repository System. 2009, 9, 23-31.  http://www.crl.nitech.ac.jp/ar/2009/2331_crl_ar2009_review.pdf

2.g-Al2O3

Langer V., Křestan J., Smrčok L. Gamma-alumina: a single-crystal X-ray diffraction study Acta Crystallographica C 2006, 62, 9, i83-i84. https://doi.org/10.1107/S0108270106026850.

Samain L., Jaworski A., Edén M., Ladd D.M., Seo D.-K., Garcia-Garcia F. J., Häussermann U. Structural analysis of highly porous γ-Al2O3. Journal of Solid State Chemistry 2014, 217, 1-8. https://doi.org/10.1016/j.jssc.2014.05.004.

Pinto H.P., Nieminen R.M., Elliott S.D. Ab initio study of gamma-Al2O3 surfaces. Physical Review B 2004, 70, 12, 125402/1-11. ISSN 1550-235X (electronic).

DOI: 10.1103/physrevb.70.125402.

3.Al2O3×SiO2  (sillimanite)

Burnham C.  Refinement of the crystal structure of sillimanite* Zeitschrift für Kristallographie, vol. 118, no. 1-2, 1963, pp. 127-148.  https://doi.org/10.1524/zkri.1963.118.1-2.127.

Durović S. A statistical model for the crystal structure of mullite. Soviet Physics - Crystallography, 1962, 7, 3, 271-278. https://rruff.info/uploads/SPC7_271.pdf.

Agrell S. O., Smith J. V. X-ray crystallography of mullite, sillimanite and praguite. Acta Crystallographica 1957, 10, 761-761.
  1. 3Al2O3×2SiO2 (mullite)

Agrell S. O., Smith J. V. X-ray crystallography of mullite, sillimanite and praguite. Acta Crystallographica 1957, 10, 12, 761-761.

  1. SiO2 (quartz)

Glinnemann J., King H. E., Jr., Schulz H., Hahn Th., La Placa S. J., Dacol F. Crystal structures of the low-temperature quartz-type phases of SiO2 and GeO2 at elevated preassure Zeitschrifh fur Kristallographic 1992, 198, 177-212. by R. Oldenbourg Verlag. Munchen 1992 - 0044-2968. https://rruff-2.geo.arizona.edu/uploads/ZK198_177.pdf.

14) In the tables, the lattice constants should be small letters and you must include unit of measurement.

This remark of the reviewer was taken into account.

15) Lines 272-273. The explanation can not be correct. First, the penetration depth of Cu Kalpha X-rays in Al2O3 is just few tens of um, that is much smaller than the coating thickness. Second, the inclusion of Si, at which the growth stops, is Si, not SiO2 (form EDX maps).

As a comment to the reviewer's note, the authors assured that the penetration depth of X-rays depends on the material under study and in the case of PEO layers can exceed 50-100 microns. In particular, it was previously shown that the thickness of the PEO layer after an hour of synthesis reached approximately 80…100 µm.  At the same time, X-ray diffraction analysis of these layers showed the presence of rather intense reflections of aluminum (the main element in the substrate for the synthesis of PEO layers) in the diffraction patterns. On this basis, it was believed that X-rays pass through a PEO layer even more than 80–100 µm thick.

15) How did you measure the thickness of such inhomogeneous layers? Also, in line 247: “increase in the number of microcracks and pores”, how did you evaluate the number of cracks and pores?

The authors have analyzed the reviewer's comments and provide the following explanations and clarifications. The method for measuring the thickness of PEO layers is described in the methodological part of the article. For the synthesis of PEO layers, samples with an area of 30x50 mm2 were used. The thickness was measured in different points (at least 10 times) on the surface of each PEO layer and its average value was determined. This was sufficient to provide a thickness estimate with an accuracy of ±10%.

In this article, only qualitative estimates of the number of microcracks and pores on the outer surface of the synthesized PEO layers were used. However, in a previous publication devoted to determining the corrosion resistance of a PEO layer on AK9 and AK12 alloys, when microcracks and pores are important structural characteristics, their number was estimated using a digital image processing program (developed in the Karpenko Physico Mechanical Institure of the National Academy of Sciences of Ukraine) [Ivasenko I., Posuvailo V., Klapkiv M., Vynar V., Ostap’yuk S. Express method for determining the presence of defects of the surface of oxide-ceramic coatings Materials Science. 2009, 45, № 3, 460–464. https://doi.org/10.1007/s11003-009-9191-6.]. It is shown that the addition of hydrogen peroxide to the base electrolyte increases the number of defects in the synthesized PEO layer. Therefore, the authors had no doubts that with a longer duration of synthesis, this regularity would be preserved. To illustrate this conclusion, the authors added additional photographs of the outer surface of the synthesized PEO layers to the text of the article. They make it possible to assess the validity of the conclusion that the number of defects increases when hydrogen peroxide is added to the electrolyte.

16) Figure 13. The thickness is identical within the experimental error. There is nothing to discuss.

To ensure high abrasive wear resistance and durability (for example, by synthesizing PEO layers on pulleys of combines operated in the presence of an abrasive in the form of sand), it is necessary to form PEO layers with a thickness of 150–200 µm. The process of synthesizing PEO layers is rather energy-intensive. However, the treatment of thousands of the pulleys in an electrolyte with the addition of hydrogen peroxide will provide tangible energy savings for their surface hardening. Indeed, an increase in the thickness of the PEO layers on the pulleys even by 5–10% (through the addition of H2O2 to the electrolyte) will give tangible energy savings (for the synthesis of a layer of unit thickness) and increase the durability of the pulleys, and hence the serviceability of the combines as a whole.

Reviewer 7 Report

Coatings-2165386

Authors have done the part of corrections and suggestions. However, some minor corrections and explanation must be done previous to publish the paper in the Coatings. Specific questions and comments are listed as follow

The author only present cross-sectional image of PEO layer synthesized in 3 g/l KOH + 2 g/l Na2SiO3 + 3 g/l H2O2 electrolyte, PEO layer synthesized in alkaline electrolyte of 3 g/l KOH + 2 g/l Na2SiO3 composition and PEO layer synthesized in 224 g/l KOH + 2 g/l Na2SiO3 electrolyte, and PEO layer synthesized in 3 g/l KOH + 2 g/l Na2SiO3 + 3g/l H2O2 composition. However, no SEM image regarding surface morphology of these 4 various PEO coated layer was observed. I suggest that the authors present the surface morphology of these 4 types of PEO coated layer.

8- Some references about coating may be useful for this Article: Coatings 2021, 11 (7), 747 and Materials 2022, 15 (23), 8300.

Author Response

Answers to the second reviewer:

1) Authors have done the part of corrections and suggestions. However, some minor corrections and explanation must be done previous to publish the paper in the Coatings. Specific questions and comments are listed as follow

The author only present cross-sectional image of PEO layer synthesized in 3 g/l KOH + 2 g/l Na2SiO3 + 3 g/l H2O2 electrolyte, PEO layer synthesized in alkaline electrolyte of 3 g/l KOH + 2 g/l Na2SiO3 composition and PEO layer synthesized in 224 g/l KOH + 2 g/l Na2SiO3 electrolyte, and PEO layer synthesized in 3 g/l KOH + 2 g/l Na2SiO3 + 3g/l H2O2 composition. However, no SEM image regarding surface morphology of these 4 various PEO coated layer was observed. I suggest that the authors present the surface morphology of these 4 types of PEO coated layer.

The reviewer's comment and wishes were taken into account. SEM images have been added to the text of the article. They demonstrate the surface morphology of 4 different PEO layers synthesized on the surface of two aluminum alloys (AK9 and AK12) in a base electrolyte and with the addition of hydrogen peroxide to it.

2) Some references about coating may be useful for this Article: Coatings 2021, 11 (7), 747 and Materials 2022, 15 (23), 8300.

The authors took into account the advice of the reviewer and used the references recommended by him in the introduction to the article to expand the scope of the PEO technology. Namele:

  1. Saberi, A., Bakhsheshi-Rad, H. R., Abazari, S., Ismail A. F.,  Sharif  S.,  Ramakrishna S., Daroonparvar M.,  Berto F. A Comprehensive review on surface modifications of biodegradable magnesium-based implant alloy: polymer coatings opportunities and challenges Coatings 202111(7), 747; https://doi.org/10.3390/coatings11070747.
  2. Bakhtiari-Zamani H., Saebnoori E., Bakhsheshi-Rad H. R., Berto F. Corrosion and wear behavior of TiO2/TiN duplex coatings on titanium by plasma electrolytic oxidation and gas nitriding Materials2022, 15(23), 8300; https://doi.org/10.3390/ma15238300.

Round 2

Reviewer 2 Report

The authors revised manuscript and introduced significant corrections. However, some parts of manuscript should be modified.

In the revised work the authors replicated the experiment using the same electrolyte composition applied in  [Kovalchuk I.V., Yurkevych R.M., Posuvailo V.M. Crystal structure of oxide ceramic coatings obtained on alloys with a high silicon content. Physicochemical Mechanics of Materials. 2022, Vol. 58. P. 50-57] . PEO layer was deposited on Al alloys containing silicon.

Due to this reason, the authors should strongly point out all differences between the actual and the former paper to demonstrate the novelty and importance experiment presented in this manuscript.

In reply to p.9 in my first review, the authors claimed that: “We did not determine the thickness of the PEO layers. Increasing silicon content reduces the thickness of the PEO layers, we investigated this in our previous work [Kovalchuk I.V., Yurkevych R.M., Posuvailo V.M. Crystal structure of oxide ceramic coatings obtained on alloys with a high silicon content. Physicochemical Mechanics of Materials. 2022, Vol. 58. P. 50-57]. “

Why did the authors not determine the thickness of PEO layer as was done in the previous work?  [Physicochemical Mechanics of Materials. 2022, Vol. 58. P. 50-57]

A simple estimation of the thickness of PEO layer can be provided from the analysis SEM images.

The choice of current density and the length of pulses should be explained. If it was examined in a separate paper appropriate references should be provided. 

Reviewer 4 Report

Some improvement were adopted. Based on this, it seems that the revised manuscript deserves its final publication.

Author Response

We would like to thank the Reviewers for the valuable remarks to our manuscript. We have revised the whole paper as recommended in the revision. The changes in the manuscript are written in blue. You will find our answers and comments (text in red colour) to the Reviewers question bellow.

Round 3

Reviewer 2 Report

I recommend this manuscript in this form for publication in this journal.

Author Response

Dear reviewer. Thank you for your valuable comments on our article. They allowed to make it better.